# Learning to Attack Federated Learning: A Model-based Reinforcement Learning Attack Framework

**Henger Li**,* **Xiaolin Sun**,* **and Zizhan Zheng**
Department of Computer Science
Tulane University
New Orleans, LA 70118
{hli30, xsun12, zzheng3}@tulane.edu

## Abstract

We propose a model-based reinforcement learning framework to derive untargeted poisoning attacks against federated learning (FL) systems. Our framework first approximates the distribution of the clients' aggregated data using model updates from the server. The learned distribution is then used to build a simulator of the FL environment, which is utilized to learn an adaptive attack policy through reinforcement learning. Our framework is capable of learning strong attacks automatically even when the server adopts a robust aggregation rule. We further derive an upper bound on the attacker's performance loss due to inaccurate distribution estimation. Experimental results on real-world datasets demonstrate that the proposed attack framework significantly outperforms state-of-the-art poisoning attacks. This indicates the importance of developing adaptive defenses for FL systems.

## 1 Introduction

Federated learning (FL) is a promising machine learning framework that allows multiple devices with private data to jointly train a learning model (coordinated by a server) without sharing their local data. It has recently been applied to consumer digital products [34], credit risk prediction [1], drug discovery [35], and digital health [40]. However, federated learning systems are vulnerable to adversarial attacks [32] such as model poisoning attacks [7, 52, 15, 4], data poisoning attacks [5, 18, 20], and inference attacks [36, 22, 60]. To this end, various robust aggregation rules such as coordinate-wise median [54], trimmed mean [54], Krum [8], norm clipping [48], geometric median [38], and FLTrust [10] have been proposed. However, these defenses are mainly evaluated against manually crafted myopic attack policies [44]. Their robustness in the face of advanced attacks remains unknown.

Due to the distributed nature of FL systems, a malicious device typically has limited knowledge about benign devices and system dynamics. To fully reveal the vulnerabilities of FL systems, it is therefore crucial to develop strong attacks that can best utilize the limited global knowledge. In this work, we take a first step in this direction by considering the white-box attack setting where the attacker has some global knowledge about the FL system and the server's algorithm, but has no access to the private data of benign devices, a reasonable assumption for real-world FL systems. To derive strong adaptive attacks, we propose to leverage the power of model-based reinforcement learning (RL) by integrating *distribution learning* and *policy learning*. A key observation of our approach is that although accurate information about individual devices can be hard to obtain in FL, it is often possible to infer their aggregated data distribution from publicly available model updates, which

---

*Equal contribution.

36th Conference on Neural Information Processing Systems (NeurIPS 2022).

is sufficient to derive strong attacks. In particular, the set of malicious devices first cooperatively estimate the aggregated data distribution through gradient inversion [22, 60]. The learned distribution is then used to build a simulator of the FL environment, which is utilized to derive an adaptive attack policy through reinforcement learning. We focus on untargeted model poisoning in this work, where the malicious devices aim to reduce the accuracy of the global model as much as possible by sending crafted gradient information to the server. However, our proposed framework can potentially be applied to other types of attacks in both the white-box and the more challenging black-box settings.

Our model-based approach distinguishes from existing work on reinforcement learning based adversarial attacks against machine learning systems [47, 58, 59]. In particular, we consider a more realistic threat model where the attacker might not always be selected due to subsampling nor does it have prior information about the distribution of the aggregated data. The attackers need to efficiently learn the distribution along the federated learning process in real time. In contrast, previous works typically assume more powerful attackers that can attack at any time and have full knowledge about the environment. Thus, they typically adopt a purely model-free approach, which is infeasible in attacking FL systems due to the large number of samples needed to be effective.

**Our contributions.** We advance the state-of-the-art in the following aspects. First, we propose a novel reinforcement learning attack framework against federated learning systems by integrating distribution learning and policy learning. Second, we theoretically quantify the effect of inaccurate distribution learning and heterogeneous local data distributions on the optimal attack performance. Third, our experiments on real-world datasets demonstrate that our RL-based attack method consistently outperforms existing model poisoning attacks [7, 15, 52] and significantly reduces the global model accuracy even when robust aggregation rules are applied. These findings indicate the importance of developing adaptive defenses for FL systems.

## 2    Related Work

**Poisoning attacks and defenses.** To compromise the integrity of the target model in federated learning, both targeted poisoning attacks [7, 4, 5] that aim to misclassify a specific set of inputs, and untargeted attacks [15, 52, 43] aiming to reduce the global model accuracy have been proposed. Existing approaches typically adopt a heuristics-based method [52]) or optimize a myopic goal [15, 43]), and are usually sub-optimal, especially when a robust aggregation rule is adopted. Further, they often require access to benign agents' local updates or the accurate global model parameters of the next round [52, 15]) to make significant attack impact. In contrast, our reinforcement learning based attack requires less global knowledge and targets a long-term attack goal.

Various defenses have been proposed for model poisoning attacks including robust-aggregation-based approaches and detection-based approaches. The former includes dimension-wise filtering that considers each dimension of local updates separately [6, 54], client-wise filtering that aims to restrict or even remove the impact of potentially malicious clients [8, 8, 38, 48], and approaches that require the server to have access to a small amount of root data [10]. Our RL-based attack is effective against all these defenses. In addition, time-coupled robust aggregation methods [2, 3, 25] that target adaptive attacks and anomaly detection-based defenses [28] have been proposed recently. Our approach can potentially be extended to compromise them by explicitly encoding the history information into states or utilizing a recurrent structure.

**RL-based attacks.** Reinforcement learning has recently been utilized for developing strong attacks in various settings, including corrupting training data of online supervised and unsupervised learning [59], manipulating the combinatorial structure of graph data [13], and injecting malicious nodes into a graph [47]. RL-based attacks have also been developed to damage the performance of reinforcement learning itself, by perturbing the reward signals during the training stage [58] and corrupting the state signals received by an agent during the testing stage [57, 45]. However, these methods typically assume that the attacker has access to an accurate MDP model or simulator and has unlimited time for training the attack policy. In contrast, our method first builds a world model by learning a data distribution from the FL model updates and then constructs an approximate simulator for training our attacks. Both distribution learning and policy learning happen when FL training is ongoing. Further, previous work has mainly focused on attacking a single RL agent by an external agent rather than an insider attack in a distributed learning environment as we consider in this work.

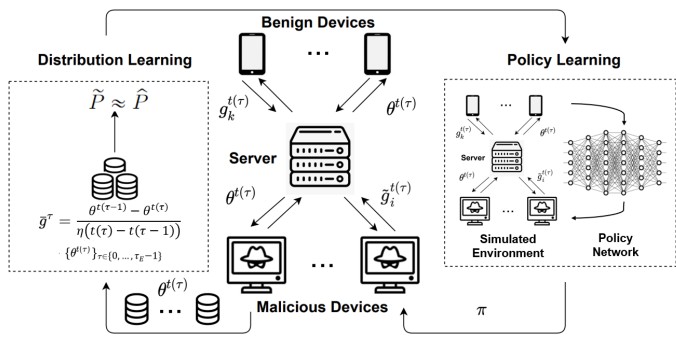

Figure 1: An overview of the RL-based attack framework against federated learning.

## 3 Approach Overview

In this section, we describe the federated learning setting considered in this work, the threat model, and the proposed RL-based attack framework.

**Federated learning.** We consider an FL setting that is similar to *federated averaging (FedAvg)* [33]. The FL system consists of a server and $K$ workers (also known as devices or clients) in which each worker has some private data. Let $[K] = \{1, 2, ..., K\}$ denote the set of workers. Coordinated by the server, the set of workers cooperate to train a machine learning model within $\mathcal{T}$ epochs by solving the following problem: $\min_{\theta} f(\theta)$ where $f(\theta) := \sum_{k=1}^{K} p_k F_k(\theta)$. Here $F_k(\cdot)$ is the local objective of worker $k$ and $p_k$ is the weight assigned to worker $k$ and satisfies $p_k \geqslant 0$ and $\sum_k p_k = 1$. The local objective $F_k(\theta)$ is usually defined as the empirical risk over worker $k$'s local data with model parameter $\theta \in \Theta$. That is, $F_k(\theta) = \frac{1}{N_k} \sum_{j=1}^{N_k} \ell(\theta; z_{jk})$, where $N_k$ is the number of data samples available locally on worker $k$, $\ell(\cdot; \cdot)$ is the loss function, and $z_{jk} := (x_{jk}, y_{jk})$ is the $j$th data sample that is drawn *i.i.d.* from some distribution $P_k$. It is typical to set $p_k = \frac{N_k}{N}$, where $N = \sum_k N_k$ is the total number of data samples across workers.

If all the workers' local data distributions are the same (i.e., $P_k = P_{k'}$ for all $k, k' \in [K]$), we call the workers' data are *i.i.d.*; otherwise, the data are *non-i.i.d.*. We write $\widehat{P}_k$ as the empirical distribution of the $N_k$ data samples drawn from $P_k$, and let $\widehat{P} := \sum_{k=1}^{K} \frac{N_k}{N} \widehat{P}_k$ denote the mixture empirical distribution across workers.

The FL algorithm (see Algorithm 1 in Appendix B) works as follows: at each time step $t \geqslant 0$, a random subset $\mathcal{S}^t$ of size $w$ is uniformly sampled without replacement from the workers set $[K]$ by the server for synchronous aggregation [30]. The process of selecting workers for aggregation is called *subsampling*. Let $\kappa = w/N$ denote the *subsampling rate*, i.e., the ratio of the selected workers number $w$ to the total number of workers $K$. Each selected worker $k \in [w]$ then samples a minibatch $b_k$ of size $B$ from its local data distribution $\widehat{P}_k$. The worker then calculates the average local gradient $g_k^{t+1} \leftarrow \frac{1}{B} \sum_{z \in b_k} \nabla_\theta \ell(\theta^t; z)$ and sends the gradient to the server. The server then uses an aggregation rule to compute the aggregated gradient $g^{t+1} \leftarrow Aggr(g_{k_1}^{t+1}, ..., g_{k_w}^{t+1})$ where $k_i \in \mathcal{S}^t$, and updates the global model parameters $\theta^{t+1} \leftarrow \theta^t - \eta g^{t+1}$ where $\eta$ is the learning rate. The newly updated model parameters $\theta^{t+1}$ is then sent to the selected workers to perform another FL iteration.

**Threat Model.** We assume that among the $K$ workers, $M(1 \leqslant M < K)$ of them are malicious. Let $\mathcal{A}$ denote the set of malicious workers. They are coordinated either by one leading attacker or an external agent. We refer such agent as a *leader agent*. These attackers are assumed to be *fully cooperative* and share the same goal of compromising the FL system. We consider untargeted model poisoning attacks in this work where the $M$ cooperative attackers send crafted local updates $\{\tilde{g}_k^t\}_{k \in \mathcal{A}}$ to the server in order to maximize the empirical loss $f(\theta)$. the batch size $B$), and their local data distributions $\{\widehat{P}_k\}_{k \in \mathcal{A}}$ (but not the benign workers' local data distributions). We further assume that the attackers obtain information about the server's training algorithm (i.e., the white-box attack setting). This information includes the server's learning rate $\eta$, the subsampling rate $\kappa$, the total number of workers $K$, the aggregation rule $Aggr$, and the total number of training epochs $\mathcal{T}$.

**RL-based online attack framework.** Our attack framework consists of the following three phases.

- **Distribution learning**: The malicious workers first jointly learn an approximation of $\widetilde{P}$ from the model updates $\{\theta^t\}$ received from the server, using a gradient inversion based inference attack [19].
- **Policy learning**: The leader agent then builds a simulator of the FL environment using the attackers' local data and the learned distribution. An optimal attack policy is then derived through reinforcement learning using data sampled from the simulator. Note that policy learning can start together with distribution learning and continue after a reasonable distribution is learned.
- **Attack execution**: The learned policy is distributed to all the malicious workers to generate attack actions. Note that attack execution can start once an initial policy is learned, which can be updated during attack execution.

We note that all the three phases happen while the federated learning process is ongoing, thus the lengths of these phases are important hyperparameters to be determined. For example, with more observations, an attacker can learn a more accurate distribution, which will help obtain a better attack policy. However, when the total time window available to attacks is limited, a longer distribution learning phase reduces the attack opportunities in Phase 3. Compared with a purely model-free approach, our model-based approach is more sample efficient, which is especially important for federated learning as a malicious worker can only attack when it is sampled by the server.

## 4 Model-based Reinforcement Learning Attack Framework

In this section, we first formulate the model poisoning attack problem as a Markov decision process (MDP). We then discuss our model-based reinforcement learning attack framework in more details.

### 4.1 Attackers' problem as a Markov decision process

The attackers' problem is formulated as an undiscounted MDP, denoted by $\mathcal{M} = (S, \boldsymbol{A}, T, r, H)$, where

- $S$ is the state space. Let $\tau \in \{0, 1, ..., H-1\}$ denote the index of the attack step and $t(\tau) \in \{0, 1, ..., \mathcal{T}-1\}$ the corresponding FL epoch when at least one attacker is selected by the server. The state at step $\tau$ is defined as $s^\tau := (\theta^{t(\tau)}, \mathcal{A}^{t(\tau)})$ where $\mathcal{A}^{t(\tau)}$ is the set of attackers selected at time $t(\tau)$, which is shared between all malicious workers.
- $\boldsymbol{A} = A^M$ is the space of the attackers' joint actions where each attacker shares the same action space $A$. If attacker $i$ is selected at $t(\tau)$, its action $a_i^\tau := \widetilde{g}_i^{t(\tau)+1} \in \mathbb{R}^d$ is the local update that attacker $i$ sends to the server at time step $t(\tau)$, where $d$ is the dimension of the model parameters. The only action available to an attacker not selected at $t(\tau)$ is $\perp$, indicating that the attacker does not send any information in that step.
- $T : S \times \boldsymbol{A} \rightarrow \mathcal{P}(S)$ is the state transition function that represents the probability of reaching a state $s' \in S$ from the current state $s \in S$ when attackers choose actions $a_1^\tau, ..., a_M^\tau$, respectively.
- $r : S \times \boldsymbol{A} \times S \rightarrow \mathbb{R}_{\geqslant 0}$ is the reward function. We define the reward at step $\tau$ as $r^\tau := f(\theta^{t(\tau+1)}) - f(\theta^{t(\tau)})$, which is determined by the current state, the next state, and the joint attack actions and is shared by all the attackers.
- $H$ is the number of attack steps in each episode and we have $t(H-1) < \mathcal{T}$.

The attackers' goal is to find a joint attack policy $\pi = (\pi_1, ..., \pi_M)$ that maximizes the expected total rewards over $H$ attack steps, i.e., $\mathbb{E}[\sum_{\tau=0}^{H-1} r^\tau]$, where $\pi_i : S \rightarrow \mathcal{P}(A)$ denotes a stationary policy of attacker $i$ that maps the state to a probability measure over $A$. Using the definition of $r^\tau$, this objective is equivalent to finding a joint policy $\pi$ that maximizes $\mathbb{E}_{\theta^{t(H)}}[f(\theta^{t(H)})]$.

A key obstacle to solving the MDP is that both the transition probabilities $T$ and the reward function $r$ depend on the joint empirical distribution across workers $\{\widehat{P}_k\}_{k \in [K]}$, which is fixed but unknown to the attackers. Although model-free reinforcement learning can bypass this difficulty, it requires a large number of samples to be effective, which is infeasible in the online attacking scenario we consider. We therefore adopt model-based reinforcement learning as a principled approach for designing adaptive attacks in the online setting. An important observation is that although the joint empirical distribution $\{\widehat{P}_k\}_{k \in [K]}$ is unknown, the attackers can learn an approximation of the mixture

distribution $\widehat{P} = \sum_{k=1}^{K} \frac{N_k}{N} \widehat{P}_k$, denoted by $\widetilde{P}$, from model updates shared by the server, which is often sufficient to simulate the behavior of benign agents and the server by assuming that each benign agent samples data from $\widetilde{P}$. This gives rise to a new MDP $\widetilde{\mathcal{M}} = (S, \boldsymbol{A}, T', r', H)$ where $T'$ and $r'$ are derived from $\widetilde{P}$. Thus, our proposed model-based reinforcement learning attack framework naturally consists of the distribution learning, policy learning, and attack execution phases.

## 4.2 Distribution learning

Initially, the attackers do not perform model-poisoning attacks. Instead, they jointly learn a mixture distribution $\widetilde{P}$ from the model updates $\{\theta^t\}$ using a gradient inversion based inference attack [19, 60]. Various gradient inversion attacks have been proposed in the literature. In this work, we adapt the *inverting gradients* (IG) method [19] to distribution learning. The IG method reconstructs data samples by optimizing a loss function based on the angle (i.e., cosine similarity) between the gradient generated from true data and that from the reconstructed data. The primary goal of IG is to reconstruct the original data samples, which is more ambitious than what the attackers need in our setting. On the other hand, recent works on gradient inversion including IG have focused on the server side, where the true gradients of each individual worker can be easily obtained from model updates. In contrast, the attackers only obtain approximated and aggregated gradient information

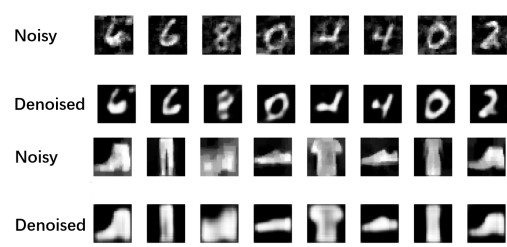

Figure 2: Examples of reconstructed images (before and after denoising) for MNIST (upper) and Fashion-MNIST (lower) datasets.

from consecutive model updates received from the server, due to model aggregation and subsampling. Despite these differences, our experiment results show that the $\widetilde{P}$ learned using IG can help derive an effective attacker policy (see Figure 4(c)).

As shown in Algorithm 2 in Appendix B, for each epoch $t(\tau)$ that at least one attacker is selected, the leader agent obtains the model update from one of the attackers and calculates the batch-level gradient as $\bar{g}^\tau := (\theta^{t(\tau-1)} - \theta^{t(\tau)})/(\eta(t(\tau) - t(\tau-1)))$. The leader agent then starts with a batch of (randomly generated) dummy data and dummy labels $D_{dummy}$, which is updated iteratively by solving the following optimization problem: $\arg\min_{D_{dummy}} 1 - \cos(\nabla_\theta F_{dummy}(\theta^{t(\tau)}), \bar{g}^\tau) + \frac{\beta}{B'} \sum_{(x,y) \in D_{dummy}} \mathrm{TV}(x)$, where $\cos(A, B) := \frac{\langle A, B \rangle}{\|A\| \|B\|}$ is the cosine similarity between two vectors $A$ and $B$, $F_{dummy}(\theta) = \frac{1}{B'} \sum_{(x,y) \in D_{dummy}} \ell(\theta; (x, y))$, $B'$ is the number of reconstructed data per epoch (the size of the dummy data batch), $\mathrm{TV}(x)$ is the total variation [41] of $x$, and $\beta$ is a fixed parameter. The process terminates after $max\_iter$ iterations, then outputs the updated data as the reconstructed data samples. We observed that although the data samples generated by IG resemble true samples, they contain a certain amount of noise as shown in Figure 2, making the learned distribution less representative of the true distribution. To reduce noise, we adapt the method of denoising auto-encoder [50]. We utilize the clean data owned by attackers and add Gaussian noise to them to simulate the noise in the reconstructed data samples. The clean data and the synthetic noisy data are then paired to train an autoencoder for denoising, which is then used to remove the noise in reconstructed data samples as shown in the figure. The approximated mixture distribution $\widetilde{P}(\tau)$ consists of the reconstructed data up to $t(\tau)$ and the $M$ attackers' local data. The distribution learning phase starts at the first FL epoch and continues for $\tau_E$ steps. The learned distribution is shared with all the attackers.

Although we adopt the IG method in this work due to its simplicity, other more recent approaches such as the GradInversion method [55] and gradient inversion with a trained generative model [23] can be easily incorporated into our framework, which can potentially learn $\widetilde{P}$ in more challenging settings for complex datasets like ImageNet [14], deep networks, and large batch sizes. On the other hand, we show in the experiments that our RL-based attack trained using attackers' local data only is still effective and surpasses all the baselines, while distribution learning further boosts the attack performance. A detailed discussion on gradient inversion attacks and defenses is provided in Appendix C

## 4.3 Policy learning

Once the leader agent obtains the approximated distribution $\widetilde{P}$, it can simulate the behavior of the server and that of normal workers. In particular, to simulate the behavior of a normal worker in each FL training step, a minibatch of size $B$ is $i.i.d.$ sampled from $\widetilde{P}$. With experiences sampled from the simulated environment, the leader agent can learn a joint attack policy that maximizes the empirical loss using a state-of-the-art (deep) reinforcement learning algorithm, e.g., TD3 [17] or PPO [42].

Note that the leader agent does not need to wait until a good distribution has been learned to start training the policy. Instead, policy learning can start together with distribution learning. Initially, the leader agent uses the attackers' local data to train the policy, which will be continuously updated while more data samples are being generated. To reduce the training overhead, we assume that all the malicious workers share the same attack policy (i.e., $\pi_1 = \pi_2 = \cdots = \pi_M$) in this work, which achieves good attack performance in our experiments. Extension to general joint policies will be considered in our future work.

When we train a small neural network with federated learning, it is natural to use $(\theta^{t(\tau)}, \mathcal{A}^{t(\tau)})$ as the state, and the crafted gradient $\widetilde{g}^{t(\tau)}$ as the action. When we use the federated learning system to train a large neural network, however, this approach does not scale as it results in an extremely large search space that requires both large runtime memory and long training time, which is usually prohibitive. To solve this problem, we propose to compress the state and action spaces for high-dimensional models as follows.

To compress the state space, we first observe that as the set of attackers are fully cooperative and share the same policy, there is no need to distinguish them in the state when the server does not track the behavior of individual workers. Thus, we replace $\mathcal{A}^{t(\tau)}$ by the number of attackers sampled in $t(\tau)$, denoted by $m^{t(\tau)}$. Further, instead of using the entire set of model parameters $\theta^{t(\tau)}$ in the state, the parameters of the last hidden layer of the current neural network model is used. This is because the last hidden layer passes on values to the output layer and typically carries information about important features of the model [46]. Note that the approximated state is used to define the policy only. The true state is still the full FL model that determines transition probabilities and rewards.

It is more challenging to compress the action space. We observe that a policy that manipulates the model parameters of the last hidden layer only works well for certain aggregation rules such as Krum and coordinate-wise median. However, it becomes less effective for stronger defenses such as coordinate-wise median with clipping. Given that it is challenging to identify a small subset of model parameters that can lead to most damage to model accuracy when manipulated, we adopt a different approach in this work.

The main idea is to search for a model update direction that can lead to a large empirical loss using gradient ascent, with its parameters identified by reinforcement learning. To this end, we define the local search objective as $L(\theta) := (1 - \lambda)F(\theta) + \lambda \cos(\theta^{t(\tau)} - \theta, g(\theta^{t(\tau)}))$ where $F(\theta) = \mathbb{E}_{z \sim \widetilde{P}}[\ell(\theta; z)]$ models the empirical loss and $g(\theta^{t(\tau)}) = \mathbb{E}_{z \sim \widetilde{P}}[\nabla_\theta \ell(\theta^{t(\tau)}; z)]$ captures the average update direction from normal devices, both are estimated from $\widetilde{P}$. The second term in $L(\theta)$ is used to control the deviation of the model update from the normal direction (measured by the cosine similarity) so that the adversarial input cannot be easily identified by the server. The parameter $\lambda \in [0, 1]$ is used to balance the loss and the deviation. To solve the problem, we start with $\theta = \theta^{t(\tau)}$ and generate $G$ trajectories, where each trajectory involves $E$ model updates. Each model update involves a single gradient ascent step using a minibatch of size $\widetilde{B}$ sampled from $\widetilde{P}$. Let $\theta_k$ denote the new model parameters found by the $k$-th trajectory after $E$ updates. The update direction is then set as $\frac{1}{G}\sum_{k=1}^{G} \theta_k$ and each attacker's action in $t(\tau)$ is computed as $\widetilde{g}^{t(\tau)+1} = \gamma(\theta^{t(\tau)} - \frac{1}{G}\sum_{k=1}^{G} \theta_k)$. The scaling factor $\gamma \geq 0$ is used to control the magnitude of the crafted gradient, which is needed as most robust aggregation rules apply a certain type of filtering rule to mitigate the effect of malicious attacks. We assume that $G$ is fixed while $\gamma, E, \lambda$ are parameters to be learned by reinforcement learning. Thus, the action space for the attackers' MDP becomes a 3-dimensional real space.

## 4.4 Attack execution

Both distribution learning and policy learning can start from the first epoch of federating learning and continue while federating learning is ongoing. The simulated environment is updated when a

new estimated distribution $\widetilde{P}$ is learned. Although the attackers may choose to start attacking during distribution learning, we observe that this can blur the gradient information and make distribution learning less accurate. Thus, we assume that each attacker starts attacking once the distribution learning phase is finished, and applies the latest learned attack policy during the remaining epochs of the federated learning process. During attack execution, each selected attacker first notifies other attackers so that every one knows the number of attackers that are sampled in that epoch. Each selected attacker then generates a crafted gradient according to the process described above with the parameters obtained from the latest learned policy.

The attackers' total training time (including distribution learning and policy learning) should be significantly less than the total FL training time so that the attackers have time to execute the attacks. In real-world FL training, the server usually must wait for some time (typically ranging from 1 minute to 10 minutes) before it receives responses from the clients [53, 9, 24]. In contrast, the leader agent does not incur such time cost in training attackers' policies using a simulated FL environment. Therefore, an epoch in policy learning is typically much shorter than an FL epoch, making it possible to train the attack policy with a large number of episodes. In addition, the leader agent is usually equipped with GPUs, or other parallel computing facilities and can run multiple training episodes in parallel [11]. We compare the actual running time of our RL-based attack against different defenses in our experiment setting in Appendix E.2.

## 5 Impact of Inaccurate Distribution Learning and Data Heterogeneity

Our model-based RL attack employs the estimated data distribution $\widetilde{P}$ to simulate the behavior of benign workers, which can suffer from two types of errors. First, $\widetilde{P}$ can be far away from the true mixture distribution $\widehat{P}$ due to inaccurate distribution learning. Second, benign workers may vary in their local data distributions $\widehat{P}_k$, which cannot be fully captured by a single mixture distribution. In this section, we study how the attack performance is affected by these two factors, which provides insights into properly distributing resources between the three phases of our attack method.

Our analysis is adapted from recent works that study the impact of model inaccuracy on the performance of model-based reinforcement learning [56, 31, 58] by addressing two new challenges. First, we need to establish the connection between the inaccuracy in data distribution $\widetilde{P}$ and the inaccuracy in the corresponding MDP as both the reward function and the transition dynamics depend on $\widetilde{P}$. Second, although there are different ways to measure the distance between two models [56], it makes more sense to use the 1-Wasserstein distance [49] to measure the distance between two data distributions. This, however, requires bounding the Lipschitz constant of the optimal value function [56]. Although this is a challenging task for general RL tasks, we are able to show that this is indeed the case in our setting under the following assumptions. The first assumption models the inaccuracy of distribution learning as well as the heterogeneity of benign workers' local data.

**Assumption 1.** $W_1(\widetilde{P}, \widehat{P}_k) \leqslant \delta$ *for any benign worker* $k$, *where* $W_1(\cdot, \cdot)$ *is the 1-Wasserstein distance [49].*

We further need the following standard assumptions on the loss function.

**Assumption 2.** *Let* $Z$ *denote the domain of data samples across all the workers. For any* $s_1, s_2 \in S$ *and* $z_1, z_2 \in Z$, *the loss function* $\ell : S \times Z \to \mathbb{R}$ *satisfies:*

1. $|\ell(s_1, z_1) - \ell(s_2, z_2)| \leqslant L\|(s_1, z_1) - (s_2, z_2)\|_2$ *(Lipschitz continuity w.r.t. s and z);*
2. $\|\nabla_s \ell(s_1, z_1) - \nabla_s \ell(s_1, z_2)\|_2 \leqslant L_z\|z_1 - z_2\|_2$ *(Lipschitz smoothness w.r.t. z);*
3. $\ell(s_2, z_1) \geqslant \ell(s_1, z_1) + \langle \nabla_s \ell(s_1, z_1), s_2 - s_1 \rangle + \frac{\alpha}{2}\|s_2 - s_1\|_2^2$ *(strong convexity w.r.t. s);*
4. $\ell(s_2, z_1) \leqslant \ell(s_1, z_1) + \langle \nabla_s \ell(s_1, z_1), s_2 - s_1 \rangle + \frac{\beta}{2}\|s_2 - s_1\|_2^2$ *(strong smoothness w.r.t. s);*
5. $\ell(\cdot, \cdot)$ *is twice continuously differentiable with respect to s.*

where $\|(s_1, z_1) - (s_2, z_2)\|_2^2 := \|s_1 - s_2\|_2^2 + \|z_1 - z_2\|_2^2$. For simplicity, we further make the following assumption on the FL environment, although our analysis can be readily applied to more general settings.

**Assumption 3.** *The server adopts FedAvg without subsampling* ($w = K$). *All workers have same amount of data* ($p_k = \frac{1}{K}$) *and the local minibatch size* $B = 1$. *In each epoch of federated learning,*

*each normal worker's local minibatch is sampled independently from the local empirical data distribution $\widehat{P}_k$.*

Since no subsampling is considered in this section, with a slight abuse of notation, we let index $t$ denote both an attack step and the corresponding FL epoch. Let $\mathcal{M} = (S, \boldsymbol{A}, T, r, H)$ denote the true MDP for the attackers, and $\widetilde{\mathcal{M}} = (S, \boldsymbol{A}, T', r', H)$ the simulated MDP when the local distribution of any benign worker is estimated as $\widetilde{P}$. The following theorem captures the attack performance loss due to inaccurate distribution learning (see Appendix D for the proof).

**Theorem 1.** *Let $\mathcal{J}_{\mathcal{M}}(\pi) := \mathbb{E}_{\pi,T,\mu_0}[\sum_{t=0}^{H-1} r(s^t, a^t, s^{t+1})]$ denote the expected return over $H$ attack steps under $\mathcal{M}$, policy $\pi$ and initial state distribution $\mu_0$. Let $\pi^*$ and $\widetilde{\pi}^*$ be the optimal policies for $\mathcal{M}$ and $\widetilde{\mathcal{M}}$ respectively, with the same initial state distribution $\mu_0$. Then,*

$$|\mathcal{J}_{\mathcal{M}}(\pi^*) - \mathcal{J}_{\mathcal{M}}(\widetilde{\pi}^*)| \leqslant 2H\epsilon\delta[(L + L_v)L_z\eta + 2L],$$

*where $\epsilon = \frac{K-M}{K}$ is the fraction of benign nodes, $L_v \leqslant \sum_{t=0}^{H-1}(K_F)^t(L + LK_F)$, and $K_F \leqslant \epsilon\max\{|1 - \eta\alpha|, |1 - \eta\beta|\}$.*

In practice, the learning rate $\eta$ is typically small enough so that $\max\{|1 - \eta\alpha|, |1 - \eta\beta|\} \leqslant 1$. In this case, $L_v$ is bounded by $\frac{L(1+K_F)}{1-K_F} \leqslant \frac{L(1+\epsilon)}{1-\epsilon}$. Therefore, we have $|\mathcal{J}_{\mathcal{M}}(\pi^*) - \mathcal{J}_{\mathcal{M}}(\widetilde{\pi}^*)| = O(H\frac{\epsilon}{1-\epsilon}\eta\delta)$. To ensure convergence, we typically have $\eta = O(\frac{1}{\sqrt{H}})$ [39], thus $|\mathcal{J}_{\mathcal{M}}(\pi^*) - \mathcal{J}_{\mathcal{M}}(\widetilde{\pi}^*)| = O(\frac{\epsilon}{1-\epsilon}\delta\sqrt{H})$. This result clearly shows how the loss of attack performance depends on the fraction of benign nodes, the inaccuracy of distribution learning, and the time horizon.

## 6  Experiments

In this section, we compare our RL-based attack with state-of-the-art model poisoning attacks on real-world datasets. Our code is available at https://github.com/SliencerX/Learning-to-Attack-Federated-Learning.

### 6.1  Experiment setup

**Datasets**. We conduct extensive experiments on four real-world datasets: MNIST [27], Fashion-MNIST [51], EMNIST [12], and CIFAR-10 [26]. Due to space limitation, experiment results for Fashion-MNIST, EMNIST, and CIFAR-10 are provided in Appendix E. For the *i.i.d.* setting, we randomly split the dataset into $K$ groups, each of which consists of the same number of training samples. For the *non-i.i.d.* setting, we follow the method of [15] to quantify the heterogeneity of local data distribution across clients. Suppose there are $C$ classes in the dataset, e.g., $C = 10$ for the MNIST and Fashion-MNIST datasets. We evenly split the worker devices into $C$ groups (with the $M$ attackers evenly distributed across the $C$ groups), where each group is assigned $1/C$ of training samples as follows. A training instance with label $c$ is assigned to the $c$-th group with probability $q \geqslant 1/C$ and to every other group with probability $(1 - q)/(C - 1)$. Within each group, instances are evenly distributed. A higher $q$ indicates a higher *non-i.i.d.* degree. We set $q = 0.5$ as the default *non-i.i.d.* degree. To demonstrate the power of distribution learning, we assume that the set of attackers share $m$ true data points sampled from the training instances assigned to them. We set $m = 200$ as the default value for MNIST.

**Baselines**. We compare our RL-based attack (RL) with no attack (NA), and the state-of-the-art model poisoning FL attack methods: explicit boosting (EB) [7], inner product manipulation (IPM) [52], and local model poisoning attack (LMP) [15]. IPM manipulates the attackers' gradients so that the inner product between the aggregation result and the true gradient is negative. This requires access to the average of normal workers' gradients in each FL epoch, which is usually unavailable in practice. LMP generates myopic attacks by solving an optimization problem in each FL epoch. In addition to the server's aggregation rule, it also requires access to normal workers' local models. Although LMP with partial knowledge is also presented in [15], it performs substantially worse than the full knowledge case when the server uses the coordinate-wise median defense. We compare the RL-based attack with the more powerful full knowledge LMP below.

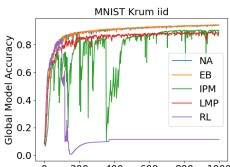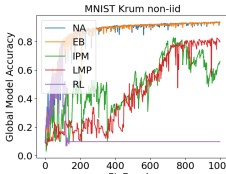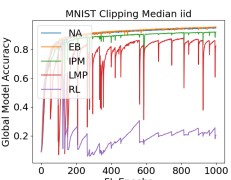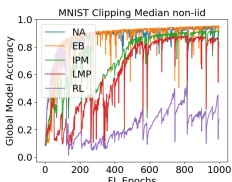

Figure 3: A comparison of global model accuracy under Krum and clipping median for both *i.i.d.* data and *non-i.i.d.* data. All parameters are set as default.

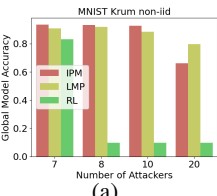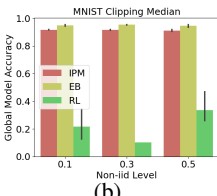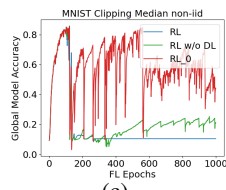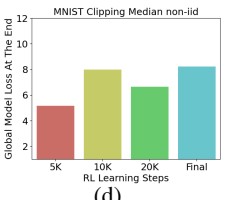

(a)                                    (b)                                    (c)                                    (d)

Figure 4: Attack performance on MNIST under (a) different number of attackers; (b) different non-iid degrees; (c) RL with and without distribution learning and $RL_0$ (zero initial data for policy learning); and (d) different policy learning lengths. Non-iid degree $q = 0.5$ in (a) and $q = 0.3$ in (c) and (d). Other parameters are set as default.

We consider four representative robust aggregation rules of different types [44]: Krum [8], geometric median [38], both of which apply client-wise filtering to model updates, coordinate-wise median [54], which adopts a dimension-wise filtering, and FLTrust [10], which requires the server to have access to a small amount of root data. In the experiments, we actually consider an extension of the vanilla coordinate-wise median where a norm clipping [48] step is first applied. This gives a more powerful defense as we observed in experiments. We set the default norm threshold to 2.

**Default FL and RL settings**. We adopt the following default parameters for the FL models: number of total workers $= 100$, number of attackers $= 20$, learning rate $\eta = 0.01$, subsampling rate $= 10\%$, the number of total FL epochs $= 1,000$. For our RL-based attack, both the distribution learning and policy learning phase start at the first FL epoch. The former ends at the 100th FL epoch when RL-based attack starts (all other attacks start at epoch 0). The policy learning phase ends at the 400th epoch. Since both the action space and state space are continuous in our setting, we choose the state-of-the-art Twin Delayed DDPG (TD3) [17] and Proximal Policy Optimization (PPO) [42] algorithms for training the attack policy in our experiments and find that TD3 gives better results in most cases. Below we report the results for TD3. We fix the initial model and the random seeds for subsampling and local data sampling for fair comparisons. See Appendix E for details of the datasets, experimental setups, and additional results.

## 6.2 Attack performance

Figure 3 shows how the test accuracy of the global model varies over FL epochs under different attacks when the server uses Krum and clipping median as the aggregation rule, respectively. Results for geometric median and FLTrust are provided in Appendix E.2. We observe that our RL-based attack performs significantly better in all the settings, despite the fact that IPM and LMP use the model updates of normal clients while RL does not. Note that for Krum, RL-based attack quickly drives the global model to a poor state ($\sim 10\%$ accuracy) once the attack starts at epoch 100 under both i.i.d. and non-i.i.d. local data distributions. Attacks become harder under clipping median due to the norm clipping but our RL-based attack still reduces global model accuracy to around $50\%$ on average. This is mainly because it targets long-term return while all other baselines are myopic. For example, in Figure 3(c), the global model accuracy drops significantly under all the attacks when five malicious devices are sampled around epoch 200. After that, the RL method keeps the accuracy at a low level, while other baselines' accuracy rebounds rapidly.

### 6.3 Ablation studies

**Impact of the number of attackers.**   Previous studies on untargeted model poisoning in federated learning typically assume a relatively large fraction of attackers. For example, the default setting is $20\%$ in [15] and $40\%$ in [52]. Figure 4(a) shows that our RL-based attack obtains superb performance even when the number of attackers (among 100 total clients) is as low as 8. In contrast, neither IPM nor LMP obtains meaningful attack performance even with 10 attackers. For 7 attackers, none of the baselines including the RL-based attack can cause significant damage to the FL system.

**Impact of non-i.i.d. degree**. Figure 4(b) shows the impact of data heterogeneity on attack performace. We use 5 different random seeds for all attacks and show the error bars. We observe that all the baselines obtain similar performance under different non-i.i.d. degrees and the impact of randomness in the testing environment on their performance is limited. On the other hand, we observe that the RL policies for $q = 0.1$ and $q = 0.5$ exhibit large variances, but even the worst-case performance of our attack outperforms the best cases of all the baselines. For $q = 0.3$, the RL-based attack can always lead to a model with a very high loss so that the model accuracy stays at a low level and is close to a constant, which explains the observed low variance in model accuracy. We expect that the variation across different RL policies is in part because the attackers always use the latest trained policy for attack execution, which does not necessarily give the best performance among all the intermediate policies trained (see also Figure 4(d)).

**Importance of distribution learning**. Figure 4(c) compares the global model accuracy of RL-based attack with distribution learning (RL) and that without distribution learning (RL w/o DL). We observe that in both cases, the model accuracy decreases dramatically after the attack starts at FL epoch 100. Further, the accuracy of RL w/o DL slightly increases up to $20\%$, while the accuracy of RL stays below $10\%$, which is consistent with our expectation that distribution learning allows the attackers to learn a better attack policy. Figure 4(c) also shows the attack performance of $RL_0$, a variant of the RL-based attack where the attackers only have 200 *unlabeled* true images used to train the denoising autoencoders, thus completely relying on distribution learning to generate labeled samples needed for policy learning. Compared with the baseline results in Figure 4(b) ($q = 0.3$), we observe that $RL_0$ still outperforms other baseline methods, further indicating the power of distribution learning. On the other hand, the fact that RL w/o DL surpasses all the baselines indicates that our approach is still applicable even when distribution learning becomes less effective in the presence of a strong defense against gradient inversion.

**Impact of training length on policy learning**. Figure 4(d) shows how the global model loss at the end of an FL training episode (in the simulated environment) varies over the RL policy training steps. We observe that longer training usually provides a better attack policy, although the training process is not stable. To fix this, one approach is to set up a separate testing environment to identify best trained policies. As mentioned above, our RL-based attack achieves promising performance even when the attackers always use the latest policy obtained during policy learning.

## 7  Conclusion

We propose a new approach for developing non-myopic attacks that can effectively compromise FL systems even with advanced defense mechanisms applied, by utilizing model-based reinforcement learning as a principled approach. While we focus on untargeted model poisoning against FL systems in this paper, our attack framework can be extended to targeted attacks (e.g., backdoor attacks) and to objectives beyond global model accuracy (e.g., fairness across clients [37, 29]). Further, our attack framework can be integrated with meta-learning [16, 21] to generalize the learned policy to different training tasks and develop black-box attacks. Another direction is to investigate novel methods to defend our adaptive attack methods. One possible solution would be to dynamically adjust FL parameters such as the subsampling rate or the aggregation rule.

## 8  Acknowledgment

This work has been funded in part by NSF grants CNS-1816495 and CNS-2146548 and Tulane University Jurist Center for Artificial Intelligence. We thank the anonymous reviewers for their valuable and insightful feedback.

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
