# Learning to Attack Federated Learning: A Model-based Reinforcement Learning Attack Framework

**Henger Li,*  Xiaolin Sun,*  and  Zizhan Zheng**
Department of Computer Science
Tulane University
New Orleans, LA 70118
{hli30, xsun12, zzheng3}@tulane.edu

# Appendix

## A   Broader Impact

To study the vulnerabilities of federated learning, we propose a model-based reinforcement learning attack framework. Our work shows that non-myopic attacks can break federated learning systems even when they are equipped with sophisticated defense rules. This reveals the urgent need of developing more advanced defense mechanisms for federated learning systems. While we have focused on adversarial attacks against federated learning in our work, we note that one possible solution to defending RL-based attacks would be to dynamically adjust FL parameters such as the subsampling rate or the aggregation rule. Future work is needed to identify how best to do so.

## B   Algorithms

Algorithm 1 gives the framework of a standard federated learning algorithm where the aggregation function, $Aggr(\cdot)$, can be either a simple average or a robust aggregation rule. Algorithm 2 gives the details of our distribution learning procedure. The algorithm first initializes $D_{reconstructed}$ with attackers' local data. A synthetic noisy dataset is built by adding Gaussian noise to $D_{reconstructed}$. A denoising autoencoder is then learned using paired clean data and noisy data. In each FL epoch, a batch of dummay data samples are first generated randomly, which are then updated iteratively by matching their average gradient with the aggregated gradient estimated from received model parameters. When no attacker is sampled in an FL epoch, the same process is applied by reusing the most recent model parameters received from the server. Due to the randomness of the algorithm, new data samples are generated and added (after denoising) to $D_{reconstructed}$ in each FL epoch during distribution learning.

## C   Discussion on Gradient Inversion Attacks and Defenses

Although federated learning is expected to protect clients' local data, it has been recently observed that sensitive information can still be inferred from the gradients or model updates shared by clients [16, 38]. In particular, it is shown in [38, 37] that using an optimization based approach, a curious server can extract both the training inputs and labels from the gradients shared by a client for a small batch size ($\leqslant 8$). This approach is further improved in [13], where it is shown that by exploiting a magnitude-invariant loss, the proposed inverting gradients (IG) method can reconstruct images in deep non-smooth architectures even in batches of 100 images. More recently,

---

*Equal contribution.

36th Conference on Neural Information Processing Systems (NeurIPS 2022).

---

**Algorithm 1** Federated Learning

---

**Input:** Initial weight $\theta^0$, $K$ workers indexed by $k$, size of subsampling $w$, local minibatch size $B$, step size $\eta$, number of global training steps $\mathcal{T}$
**Output:** $\theta^{\mathcal{T}}$
**Server executes:**
 **for** $t = 0$ to $\mathcal{T} - 1$ **do**
  $\mathcal{S}^t \leftarrow$ randomly select $w$ workers from $K$ workers
  **for** each worker $j \in \mathcal{S}^t$ **in parallel do**
   $g_j^{t+1} \leftarrow$ **WorkerUpdate**$(j, \theta^t)$
  **end for**
  $g^{t+1} \leftarrow Aggr(g_{k_1}^{t+1}, ..., g_{k_w}^{t+1}), k_i \in \mathcal{S}^t$
  $\theta^{t+1} \leftarrow \theta^t - \eta g^{t+1}$
 **end for**
**WorkerUpdate**$(j, \theta)$:
 Sample a minibatch $b$ of size $B$
 $g \leftarrow \frac{1}{B} \sum_{z \in b} \nabla_\theta \ell(\theta, z)$
 return $g$ to server

---

---

**Algorithm 2** Distribution Learning

---

**Input:** number of steps for distribution learning $\tau_E$, number of iterations for each step $max\_iter$, learning rate for FL $\eta$ learning rate for inverting gradients $\eta'$, number of reconstructed data per epoch $B'$, and model parameters $\{\theta^{t(\tau)}\}$
**Output:** $D_{reconstructed}$
$D_{Reconstructed} \leftarrow M$ attackers' local data
$D_{Noisy} \leftarrow$ Add Gaussian noise to $D_{reconstructed}$ and clip data to the valid range
Train a denoising autoencoder $A_{denoise}$ using $D_{reconstructed}$ and $D_{noisy}$
**for** $\tau = 0$ to $\tau_E$ **do**
 Generate $D_{dummy}$ with $B'$ random data and label pairs
 Compute aggregated gradient $\bar{g}^\tau \leftarrow (\theta^{t(\tau-1)} - \theta^{t(\tau)})/(\eta(t(\tau) - t(\tau - 1)))$
 **for** $i = 0$ to $max\_iter - 1$ **do**
  $F_{dummy}(\theta) \leftarrow \frac{1}{B'} \sum_{(x_j, y_j) \in D_{dummy}} \ell(\theta; (x_j, y_j))$
  $\mathcal{L} \leftarrow 1 - \frac{\langle \nabla_\theta F_{dummy}(\theta^{t(\tau)}), \bar{g}^\tau \rangle}{||\nabla_\theta F_{dummy}(\theta^{t(\tau)})|| \cdot ||\bar{g}^\tau||} + \frac{\beta}{B'} \sum_{(x_j, y_j) \in D_{dummy}} \text{TV}\,(x_j)$
  $x_j \leftarrow x_j - \eta' \nabla_{x_j} \mathcal{L}, y_j \leftarrow y_j - \eta' \nabla_{y_j} \mathcal{L}, \ \forall (x_j, y_j) \in D_{dummy}$
 **end for**
 Denoise the dummy batch $D_{dummy}$ using $A_{denoise}$ and add it to $D_{reconstructed}$
**end for**

---

the GradInversion method [35] and gradient inversion with a trained generative model [17] are capable of reconstructing individual images with high fidelity from averaging gradients even for complex datasets like ImageNet [10], deep networks, and large batch sizes. Several approaches have been proposed to counter inference attacks. This includes methods that inject a limited amount of statistical noise into model updates [1, 14] and approaches that learn to perturb data representation [27] such that the data reconstructed from the perturbed representation is dissimilar to the raw data, while FL performance is maintained. However, it is unclear if these defenses can provide sufficient protection in the face of more advanced attacks. Further, they introduce extra overhead on the client side. On the other hand, our attack framework only requires a rough estimate of the joint distribution of clients' local data and can tolerate a certain level of inaccuracy in the learned dataset, which provides the attacker with extra flexibility.

## D Proof of Theorem 1

### D.1 Preliminaries

Our theoretic analysis relies on the following definitions and results. First, we formally define the Wasserstein distance [30], which will be used to measure the distance between the estimated and true

data distributions as well as the distance between the corresponding transition dynamics introduced by different data distributions.

**Definition 1.** *(Wasserstein distance) Let $(\mathbf{M}, d)$ be a metric space and $\mathcal{P}_p(\mathbf{M})$ the set of all probability measures on $\mathbf{M}$ with finite $p^{th}$ moment, then the $p^{th}$ Wasserstein distance between two probability distributions $\mu_1$ and $\mu_2$ in $\mathcal{P}_p(\mathbf{M})$ is defined as:*

$$W_p(\mu_1, \mu_2) := \left( \inf_{j \in \mathcal{J}} \int \int d(s_1, s_2)^p j(s_1, s_2) ds_1 ds_2 \right)^{1/p}$$

*where $\mathcal{J}$ is the collection of all joint distributions $j$ on $\mathbf{M} \times \mathbf{M}$ with marginals $\mu_1$ and $\mu_2$.*

In the following, we focus on 1-Wasserstein distance and denote $W(\mu_1, \mu_2) := W_1(\mu_1, \mu_2)$. Wasserstein distance is also known as "Earth Mover's distance" and measures the minimum expected distance between two sets of points where the joint distribution is constrained to match their corresponding marginals. Compared with Kullback-Leibler (KL) divergence and Total Variation (TV) distance, Wasserstein distance is more sensitive to how far the points are from each other [4].

We will also need the following special form of Lipschitz continuity from [4].

**Definition 2.** *(Lipschitz Continuity) Given two metric spaces $(\mathbf{M}_1, d_1)$ and $(\mathbf{M}_2, d_2)$, a function $f : \mathbf{M}_1 \rightarrow \mathbf{M}_2$ is Lipschiz continuous if the Lipschiz constant, defined as*

$$K_{d_1, d_2}(f) := \sup_{s_1 \in \mathbf{M}_1, s_2 \in \mathbf{M}_2} \frac{d_2(f(s_1), f(s_2))}{d_1(s_1, s_2)}$$

*is finite. Similarly, a function $f : \mathbf{M}_1 \times A \rightarrow \mathbf{M}_2$ is uniformly Lipschitz continuous in $A$ if:*

$$K^A_{d_1, d_2}(f) := \sup_{a \in A} \sup_{s_1, s_2} \frac{d_2(f(s_1, a), f(s_2, a))}{d_1(s_1, s_2)}$$

*is finite.*

Let $\mathcal{M} = (S, A, T, r)$ be a generic MDP, where $S$ and $A$ denote the state space and the action space respectively, $T(s'|s, a)$ denotes the probability of reaching a state $s'$ from the current state $s$ and action $a$, and $r(s, a, s')$ denotes the reward given the current state $s$, action $a$, and the next state $s'$. We then introduce the concept of Lipschiz model class from [4], which allows us to represent the stochastic transition dynamics of an MDP as a distribution over a set of deterministic transitions.

**Definition 3.** *(Lipschitz model class) Given a metric state space $(S, d_S)$ and an action space $A$, let $F_g$ be a collection of functions: $F_g = \{f : S \rightarrow S\}$ distributed according to $g(f|a)$ where $a \in A$. We say that $F_g$ is a Lipschitz model class if*

$$K_F := \sup_{f \in F_g} K_{d_S, d_S}(f)$$

*is finite. We say that a transition function $T$ is induced by a Lipschitz model class $F_g$ if $T(s'|s, a) = \sum_f \mathbb{1}(f(s) = s')g(f|a)$ for any $s, s' \in S$ and $a \in A$.*

We will later show that the transition dynamics of our MDP model for attackers is induced by a Lipschitz model class.

Finally we give a formal definition of finite-horizon value functions [29].

**Definition 4.** *Given an MDP $\mathcal{M}$ and a stationary policy $\pi$, the value function of $\pi$ at time $l$ is defined as $V^\pi_{\mathcal{M}, l}(s) := \mathbb{E}_{\pi, T}[\sum_{t=l}^{H-1} r(s^t, a^t)|s^l = s]$, where $r(s, a) := \mathbb{E}_{s' \sim T(\cdot|s, a)}[r(s, a, s')]$. $V^\pi_{\mathcal{M}, l}(\cdot)$ satisfies the following backward recursion form:*

$$V^\pi_{\mathcal{M}, l}(s) = \mathbb{E}_{a \sim \pi(s)}[r(s, a) + \sum_{s' \in S} T(s'|s, a)V^\pi_{\mathcal{M}, l+1}(s')]$$

*with $V^\pi_{\mathcal{M}, H-1}(s) = \mathbb{E}_{a \sim \pi(s)}[r(s, a)]$. The optimal value function is defined as $V^*_{\mathcal{M}, l}(s) := \max_\pi V^\pi_{\mathcal{M}, l}(s)$ for any $s$.*

To analyze the impact of inaccurate transition probabilities on the value function, we also make use of the following lemmas [4].

**Lemma 1.** *Given two distributions $\mu_1$ and $\mu_2$ over states $S$, a transition function $T$ induced by a Lipschitz model class $F_g$ is uniformly Lipschitz continuous in action space $A$ with a constant:*

$$K_{W,W}^A(T) := \sup_{a \in A} \sup_{\mu_1, \mu_2} \frac{W(T(.|\mu_1, a), T(.|\mu_2, a))}{W(\mu_1, \mu_2)} \leqslant K_F$$

**Lemma 2.** *Given a Lipschiz function $f : S \to \mathbb{R}$ with constant $K_{d_S, d_{\mathbb{R}}}(f)$:*

$$K_{d_S, d_{\mathbb{R}}}^A \left( \int f(s') T(s'|s, a) ds' \right) \leqslant K_{d_S, d_{\mathbb{R}}}(f) K_{d_S, W}^A(T)$$

Below we state the assumptions needed for establishing Theorem 1. The first assumption models the inaccuracy of distribution learning as well as the heterogeneity of benign workers' local data.

**Assumption 1.** $W(\widetilde{P}, \widehat{P}_k) \leqslant \delta$ *for any benign worker $k$.*

We further need the following standard assumptions on the loss function.

**Assumption 2.** *Let $Z$ denote the domain of data samples across all the workers. For any $s_1, s_2 \in S$ and $z_1, z_2 \in Z$, the loss function $\ell : S \times Z \to \mathbb{R}$ satisfies:*

1. $|\ell(s_1, z_1) - \ell(s_2, z_2)| \leqslant L\|(s_1, z_1) - (s_2, z_2)\|_2$    *(Lipschitz continuity w.r.t. $s$ and $z$);*
2. $\|\nabla_s \ell(s_1, z_1) - \nabla_s \ell(s_1, z_2)\|_2 \leqslant L_z \|z_1 - z_2\|_2$    *(Lipschitz smoothness w.r.t. $z$);*
3. $\ell(s_2, z_1) \geqslant \ell(s_1, z_1) + \langle \nabla_s \ell(s_1, z_1), s_2 - s_1 \rangle + \frac{\alpha}{2}\|s_2 - s_1\|_2^2$   *(strong convexity w.r.t. $s$);*
4. $\ell(s_2, z_1) \leqslant \ell(s_1, z_1) + \langle \nabla_s \ell(s_1, z_1), s_2 - s_1 \rangle + \frac{\beta}{2}\|s_2 - s_1\|_2^2$   *(strong smoothness w.r.t. $s$);*
5. $\ell(\cdot, \cdot)$ *is twice continuously differentiable with respect to $s$.*

where $\|(s_1, z_1) - (s_2, z_2)\|_2^2 := \|s_1 - s_2\|_2^2 + \|z_1 - z_2\|_2^2$.

For simplicity, we further make the following assumption on the FL environment, although our analysis can be readily applied to more general settings.

**Assumption 3.** *The server adopts FedAvg without subsampling ($w = K$). All workers have same amount of data ($p_k = \frac{1}{K}$) and the local minibatch size $B = 1$. In each epoch of federated learning, each normal worker's local minibatch is sampled independently from the local empirical data distribution $\widehat{P}_k$.*

### D.2    Measuring the uncertainty: from data distributions to total returns

Let $\mathcal{M} = (S, \boldsymbol{A}, T, r, H)$ denote the true MDP for attacking the federated learning system, and $\widetilde{\mathcal{M}} = (S, \boldsymbol{A}, T', r', H)$ the estimated MDP used in the policy learning stage, where $T'$ and $r'$ are derived from the estimated joint data distribution $\{\widetilde{P}_k\}$ where $\widetilde{P}_k = \widehat{P}_k$ when $k$ is an attacker and $\widetilde{P}_k = \widetilde{P}$ otherwise. Our main goal is to compare the optimal attack performance that can be obtained from the true MDP $\mathcal{M}$ and that derived from the simulated MDP $\widetilde{\mathcal{M}}$. We will focus on understanding the impact of inaccurate data distributions (obtained from distribution learning) and assume that other system parameters are known to the attackers.

Without loss of generality, we assume that the $M$ attackers' indexes are from $K - M + 1$ to $K$. Let $[M] = \{K - M + 1, ..., K\}$ denote the set of attackers and $\epsilon = \frac{K - M}{M}$ the fraction of benign nodes. We consider the idealized setting where the $M$ attackers are perfectly coordinated by a single leading attacker. Because of these simplifications, the state $s^t$ in each epoch $t$ is completely defined by the current model parameters $\theta^t$. With a slight abuse of notation, we assume $S = \Theta$ in the following.

Let $\mathcal{J}_{\mathcal{M}}(\pi) := \mathbb{E}_{\pi, T, \mu_0}[\sum_{t=0}^{H-1} r(s^t, a^t, s^{t+1})]$ denote the expected return over $H$ attack steps under the MDP $\mathcal{M}$, policy $\pi$ and initial state distribution $\mu_0$. Let $\pi^*$ be an optimal policy of $\mathcal{M}$ that maximizes $\mathcal{J}_{\mathcal{M}}(\pi)$. Define $\mathcal{J}_{\widetilde{\mathcal{M}}}(\pi)$ similarly and let $\widetilde{\pi}^*$ be an optimal policy for $\widetilde{\mathcal{M}}$, with the same initial state distribution $\mu_0$.

Our analysis is built upon the following lemma that compares the performance of $\pi^*$ and that of $\widetilde{\pi}^*$ with respect to the true MDP $\mathcal{M}$. It extends a similar result in [36] to a finite-horizon MDP where the reward in each step depends on not only the current state and action but also the next state. Note that the lemma relies on the key assumption that both $V_{\mathcal{M}, l}^*(\cdot)$ and $V_{\widetilde{\mathcal{M}}, l}^*(\cdot)$ are $L_v$-Lipschitz continuous

(with respect to the $l_2$ norm of states) for all $l$. That is, $|V^*_{\mathcal{M},l}(s_1) - V^*_{\mathcal{M},l}(s_2)| \leqslant L_v\|s_1 - s_2\|_2$ for any $s_1, s_2 \in S$ where $L_v$ is a constant independent of $l$. A similar requirement holds for $V^*_{\widetilde{\mathcal{M}},l}(\cdot)$. Let $W(T, T') := \sup_{a \in \mathbf{A}} \sup_{s \in S} W(T(\cdot|s, a), T'(\cdot|s, a))$.

**Lemma 3.** *Assume Assumption 1 and Assumption 2.1 holds and both $V^*_{\mathcal{M},l}(\cdot)$ and $V^*_{\widetilde{\mathcal{M}},l}(\cdot)$ are $L_v$-Lipschitz continuous for all $l$. Then,*

$$|\mathcal{J}_{\mathcal{M}}(\pi^*) - \mathcal{J}_{\mathcal{M}}(\widetilde{\pi}^*)| \leqslant 2H[(L + L_v)W(T, T') + 2L\epsilon\delta]$$

*Proof.* Let $F_l$ be the expected return when $\pi^*$ is applied to $\widetilde{\mathcal{M}}$ for the first $l$ steps, then switching to $\mathcal{M}$ for $l$ to $H - 1$. That is,

$$F_l = \mathop{\mathbb{E}}_{\substack{a^t \sim \pi^*(s^t) \\ t < l: s^{t+1} \sim T'(s^t, a^t), r^t = r' \\ t \geqslant l: s^{t+1} \sim T(s^t, a^t), r^t = r}} \left[ \sum_{t=0}^{H-1} r^t(s^t, a^t, s^{t+1}) \right]$$

By the definition of $F_l$, we have $\mathcal{J}_{\mathcal{M}}(\pi^*) = F_0$ and $\mathcal{J}_{\widetilde{\mathcal{M}}}(\pi^*) = F_H$, which implies that $\mathcal{J}_{\mathcal{M}}(\pi^*) - \mathcal{J}_{\widetilde{\mathcal{M}}}(\pi^*) = \sum_{l=0}^{H-1}(F_l - F_{l+1})$. Note that

$$F_l = R_{l-1} + \mathbb{E}_{s^l, a^l \sim T', \pi^*}[\mathbb{E}_{s^{l+1} \sim T(s^l, a^l)}[r(s^l, a^l, s^{l+1}) + V^*_{\mathcal{M},l+1}(s^{l+1})]]$$

$$F_{l+1} = R_{l-1} + \mathbb{E}_{s^l, a^l \sim T', \pi^*}[\mathbb{E}_{s^{l+1} \sim T'(s^l, a^l)}[r'(s^l, a^l, s^{l+1}) + V^*_{\mathcal{M},l+1}(s^{l+1})]]$$

where $R_{l-1}$ is the expected return of the first $l - 1$ steps, which are taken with respect to $\widetilde{\mathcal{M}}$. Thus,

$$F_l - F_{l+1} = \mathbb{E}_{s^l, a^l \sim T', \pi^*}[\mathbb{E}_{s^{l+1} \sim T(s^l, a^l)}[r(s^l, a^l, s^{l+1})] - \mathbb{E}_{s^{l+1} \sim T'(s^l, a^l)}[r'(s^l, a^l, s^{l+1})]]$$
$$+ \mathbb{E}_{s^l, a^l \sim T', \pi^*}[\mathbb{E}_{s^{l+1} \sim T(s^l, a^l)}[V^*_{\mathcal{M},l+1}(s^{l+1})] - \mathbb{E}_{s^{l+1} \sim T'(s^l, a^l)}[V^*_{\mathcal{M},l+1}(s^{l+1})]]$$

Define $G^*_{\widetilde{\mathcal{M}},l}(s^l, a^l) := \mathbb{E}_{s^{l+1} \sim T(s^l, a^l)}[V^*_{\mathcal{M},l}(s^{l+1})] - \mathbb{E}_{s^{l+1} \sim T'(s^l, a^l)}[V^*_{\mathcal{M},l}(s^{l+1})]$. We have

$$\mathcal{J}_{\mathcal{M}}(\pi^*) - \mathcal{J}_{\widetilde{\mathcal{M}}}(\pi^*) = \sum_{l=0}^{H-1}(F_l - F_{l+1})$$

$$= \sum_{l=0}^{H-1} \mathbb{E}_{s^l, a^l \sim T', \pi^*}\left( \mathbb{E}_{s^{l+1} \sim T(s^l, a^l)}[r(s^l, a^l, s^{l+1})] - \mathbb{E}_{s^{l+1} \sim T'(s^l, a^l)}[r'(s^l, a^l, s^{l+1})] \right)$$

$$+ \sum_{l=0}^{H-1} \mathbb{E}_{s^l, a^l \sim T', \pi^*}[G^*_{\widetilde{\mathcal{M}},l}(s^l, a^l)]$$

$$= \sum_{l=0}^{H-1} \mathbb{E}_{s^l, a^l \sim T', \pi^*}\left( \mathbb{E}_{s^{l+1} \sim T(s^l, a^l)}[\frac{1}{K} \sum_{k=1}^{K}(\ell_k(s^{l+1}) - \ell_k(s^l))] \right.$$

$$\left. - \mathbb{E}_{s^{l+1} \sim T'(s^l, a^l)}[\frac{1}{K} \sum_{k=1}^{K} \ell'_k(s^{l+1}) - \ell'_k(s^l))] \right) + \sum_{l=0}^{H-1} \mathbb{E}_{s^l, a^l \sim T', \pi^*}[G^*_{\widetilde{\mathcal{M}},l}(s^l, a^l)]$$

$$= \sum_{l=0}^{H-1} \mathbb{E}_{s^l, a^l \sim T', \pi^*}\left( \mathbb{E}_{s^{l+1} \sim T(s^l, a^l)}[\frac{1}{K} \sum_{k=1}^{K} \ell_k(s^{l+1})] - \mathbb{E}_{s^{l+1} \sim T'(s^l, a^l)}[\frac{1}{K} \sum_{k=1}^{K} \ell'_k(s^{l+1})] \right)$$

$$+ \sum_{l=0}^{H-1} \mathbb{E}_{s^l, a^l \sim T', \pi^*}\left( \frac{1}{K} \sum_{k=1}^{K} \ell'_k(s^l) - \frac{1}{K} \sum_{k=1}^{K} \ell_k(s^l) \right)$$

$$+ \sum_{l=0}^{H-1} \mathbb{E}_{s^l, a^l \sim T', \pi^*}[G^*_{\widetilde{\mathcal{M}},l}(s^l, a^l)]$$

where $\ell_k(s) := \mathbb{E}_{z_k \sim \widehat{P}_k}[\ell(s, z_k)]$, $\ell'_k(s) := \mathbb{E}_{z_k \sim \widetilde{P}_k}[\ell(s, z_k)]$ and the third equality follows from the definition of reward function $r(s, a, s') = \frac{1}{K} \sum_{k=1}^{K} \ell_k(s') - \frac{1}{K} \sum_{k=1}^{K} \ell_k(s)$, and $r'(s, a, s') = \frac{1}{K} \sum_{k=1}^{K} \ell'_k(s') - \frac{1}{K} \sum_{k=1}^{K} \ell'_k(s)$.

Since $V_{\mathcal{M},l}^*$ is $L_v$-Lipschitz, we have $|G_{\widetilde{\mathcal{M}},l}^*(s,a)| \leqslant L_v W(T(s,a), T'(s,a))$ from the definition of 1-Wasserstein distance. We further have

$$\left| \frac{1}{K} \sum_{k=1}^{K} \ell_k'(s) - \frac{1}{K} \sum_{k=1}^{K} \ell_k(s) \right| \leqslant \frac{1}{K} \sum_{k=1}^{K} |\ell_k'(s) - \ell_k(s)|$$

$$= \frac{1}{K} \sum_{k=1}^{K} \left| \mathbb{E}_{z_k \sim \widetilde{P}_k} \ell_k(s, z_k) - \mathbb{E}_{z_k \sim \widehat{P}_k} \ell_k(s, z_k) \right|$$

$$\leqslant \frac{1}{K} \sum_{k=1}^{K} L W(\widetilde{P}_k, \widehat{P}_k)$$

$$\leqslant L\epsilon\delta,$$

where the second inequality follows from the definition of 1-Wasserstein distance and Assumption 2.1, and the last inequality follows from Assumption 1 and the fact that $\widetilde{P}_k = \widehat{P}_k$ for any attacker $k$. Similarly, we have

$$\left| \mathbb{E}_{s' \sim T(s,a)} \left[ \frac{1}{K} \sum_{k=1}^{K} \ell_k(s') \right] - \mathbb{E}_{s' \sim T'(s,a)} \left[ \frac{1}{K} \sum_{k=1}^{K} \ell_k'(s') \right] \right|$$

$$\leqslant \frac{1}{K} \sum_{k=1}^{K} \left| \mathbb{E}_{s' \sim T(s,a)}[\ell_k(s')] - \mathbb{E}_{s' \sim T'(s,a)}[\ell_k'(s')] \right|$$

$$= \frac{1}{K} \sum_{k=1}^{K} \left| \mathbb{E}_{s' \sim T(s,a), z_k \sim \widehat{P}_k}[\ell_k(s', z_k)] - \mathbb{E}_{s' \sim T'(s,a), z_k \sim \widetilde{P}_k}[\ell_k(s', z_k)] \right|$$

$$\leqslant L(W(T, T') + \epsilon\delta),$$

where the last inequality follows Assumption 1, Assumption 2.1, and the property of 1-Wasserstein distance with respect to product measures. Combining the above results, we have

$$\mathcal{J}_{\mathcal{M}}(\pi^*) - \mathcal{J}_{\widetilde{\mathcal{M}}}(\pi^*) \leqslant H(L_v + L)W(T, T') + 2HL\epsilon\delta.$$

A similar argument shows that

$$\mathcal{J}_{\widetilde{\mathcal{M}}}(\widetilde{\pi}^*) - \mathcal{J}_{\mathcal{M}}(\widetilde{\pi}^*) \leqslant H(L_v + L)W(T, T') + 2HL\epsilon\delta.$$

Let $U := H(L_v + L)W(T, T') + 2HL\epsilon\delta$. We have

$$\mathcal{J}_{\mathcal{M}}(\pi^*) \leqslant \mathcal{J}_{\widetilde{\mathcal{M}}}(\pi^*) + U \leqslant \mathcal{J}_{\widetilde{\mathcal{M}}}(\widetilde{\pi}^*) + U \leqslant \mathcal{J}_{\mathcal{M}}(\widetilde{\pi}^*) + 2U.$$

$\square$

As indicated in [36], an important obstacle to applying Lemma 3 to real reinforcement learning problems is to bound the Lipschitz constant $L_v$ for optimal value functions. Further, we need to bound $W(T, T')$, the 1-Wasserstein distance between two transition functions. We study these two problems in the following two subsections, respectively.

### D.3  Lipschitz constants of value functions

In this section, we show that the Lipschitz constant $L_v$ can be upper bounded for any optimal value function in our setting. We first rewrite the update of model parameters in each epoch of FedAvg as follows:

$$f_z(s, \{\tilde{g}_i\}_{i \in [M]}) := s - \eta \frac{1}{K} \left[ \sum_{k=1}^{K-M} \nabla_s \ell(s, z_k) + \sum_{k=K-M+1}^{K} \tilde{g}_k \right] \tag{1}$$

where $s$ denotes the parameters of the current global model, $z = \{z_k\}$ denotes the set of data points sampled by each worker. Note that the above equation gives the one-step *deterministic* transition when the data samples are fixed. An important observation is that the transition function $T$ is induced by a Lipschitz model class $F_g = \{f_z : z \in Z^K\}$ with $g(f_z|a)$ equal to the probability that $z$ is

sampled according to the joint distribution $\prod_{k \in [K]} \widehat{P}_k$. Similarly, $T'$ is induced by $F_{g'} = \{f_z : z \in Z^K\}$ with $g'(f_z|a)$ equal to the probability that $z$ is sampled according to the joint distribution $\widetilde{P}^{K-M} \prod_{k=K-M+1}^{K} \widehat{P}_k$. This observation allows us to apply the techniques in [4] to bound the Lipschitz constant $L_v$ of an optimal value function once we bound the Lipschitz continuity of individual $f_z$.

We first show that for any joint action $a = \{\tilde{g}_i\}_{i \in [M]}$, the deterministic transition $f_z(\cdot, a)$ is Lipschitz continuous with a Lipschitz constant $K_{d_S, d_S}(f_z(\cdot, a))$ that can be upper bounded independent of $z$.

**Lemma 4.** *Assume Assumptions 2.3, 2.4, and 2.5 hold. For any Lipschitz model class $F_g = \{f_z : z \in Z^K\}$, we have $K_F \leqslant \max\{\epsilon|1 - \eta\alpha|, \epsilon|1 - \eta\beta|\}$.*

*Proof.* It suffices to show that for any action $a$, $K_{d_S, d_S}(f_z(\cdot, a)) \leqslant \max\{\epsilon|1 - \eta\alpha|, \epsilon|1 - \eta\beta|\}$. By (1), we have for any $s_1, s_2 \in S$,

$$
\begin{aligned}
\|f_z(s_1, a) - f_z(s_2, a)\|_2 &= \left\| s_1 - \eta \frac{1}{K} \sum_{k=1}^{K-M} \nabla_s \ell(s_1, z_k) - \left(s_2 - \eta \frac{1}{K} \sum_{k=1}^{K-M} \nabla_s \ell(s_2, z_k)\right) \right\|_2 \\
&\overset{(a)}{\leqslant} \frac{1}{K} \sum_{k=1}^{K-M} \|s_1 - \eta \nabla_s \ell(s_1, z_k) - (s_2 - \eta \nabla_s \ell(s_2, z_k))\|_2 \\
&\overset{(b)}{=} \frac{1}{K} \sum_{k=1}^{K-M} \left\| \left(I - \eta \frac{\partial^2 \ell(\bar{s}, z_k)}{\partial s^2}\right)(s_1 - s_2) \right\|_2 \\
&\overset{(c)}{\leqslant} \frac{1}{K} \sum_{k=1}^{K-M} \left\| I - \eta \frac{\partial^2 \ell(\bar{s}, z_k)}{\partial s^2} \right\|_2 \|s_1 - s_2\|_2
\end{aligned}
$$

where (a) follows from the triangle inequality, (b) follows from the fact that $\ell(s, z)$ is twice continuously differentiable with respect to $s$ and the mean value theorem, where $\bar{s}$ is a point on the line segment connecting $s_1$ and $s_2$, and $I$ is the identity matrix with its dimension equal to the dimension of the model parameters, and (c) is due to the Cauchy–Schwarz inequality.

By the strong convexity and smoothness of $\ell(s, z)$ with respect to $s$, the eigenvalues of $\frac{\partial^2 \ell(\bar{s}, z_k)}{\partial s^2}$ are between $\alpha$ and $\beta$ [22]. It follows that

$$
\left\| I - \eta \frac{\partial^2 \ell(\bar{s}, z_k)}{\partial s^2} \right\|_2 \leqslant \max\{|1 - \eta\alpha|, |1 - \eta\beta|\}, \quad \forall k
$$

Therefore, for any $s_1, s_2$,

$$
\frac{\|f_z(s_1, a) - f_z(s_2, a)\|_2}{\|s_1 - s_2\|_2} \leqslant \max\{\epsilon|1 - \eta\alpha|, \epsilon|1 - \eta\beta|\}
$$

By Definition 2, we then have

$$
\begin{aligned}
K_{d_S, d_S}(f_z(\cdot, a)) &:= \sup_{s_1, s_2} \frac{\|f_z(s_1, a) - f_z(s_2, a)\|_2}{\|s_1 - s_2\|_2} \\
&\leqslant \max\{\epsilon|1 - \eta\alpha|, \epsilon|1 - \eta\beta|\}
\end{aligned}
$$

$\square$

Note that by using a small enough learning rate $\eta$, $K_F$ can be made less than 1 so that the one-step deterministic transition becomes a contraction. We next show that the optimal value function $V^*_{\mathcal{M}, l}(\cdot)$ has a bounded Lipschitz constant. Note that the bound is independent of $\mathcal{M}$; hence it also applies to $V^*_{\widetilde{\mathcal{M}}, l}(\cdot)$

**Lemma 5.** *Assume Assumptions 2.1, 2.3, 2.4, and 2.5 hold. The optimal value function $V^*_{\mathcal{M}, l}(\cdot)$ is Lipschitz continuous with a Lipschitz constant bounded by $\sum_{t=0}^{H-l-1}(K_F)^t(L + LK_F)$.*

*Proof.* The proof is adapted from the proof of Theorem 3 in [4]. Let $Q_{\mathcal{M},l}^\pi(s,a) := r(s,a) + \sum_{s'\in S} T(s'|s,a)V_{\mathcal{M},l+1}^\pi(s')$ denote the state-action value function, where $r(s,a) = \mathbb{E}_{s'\sim T(s'|s,a)}[r(s,a,s')]$. We have for the optimal state-action value function

$$Q_{\mathcal{M},l}^*(s,a) = r(s,a) + \sum_{s'\in S} T(s'|s,a)\max_{a'\in A} Q_{\mathcal{M},l+1}^*(s',a')$$

with $Q_{\mathcal{M},H-1}^*(s,a) = r(s,a)$. The Lipschitz constant of $Q_{\mathcal{M},l}^*$ is bounded by:

$$K_{d_S,d_\mathbb{R}}^A(Q_{\mathcal{M},l}^*) \leqslant K_{d_S,d_\mathbb{R}}^A(r) + K_{d_S,d_\mathbb{R}}^A\left(\sum_{s'\in S} T(s'|s,a)\max_{a'\in A} Q_{\mathcal{M},l+1}^*(s',a')\right)$$

$$\overset{(a)}{\leqslant} K_{d_S,d_\mathbb{R}}^A(r) + K_{d_S,W}^A(T)K_{d_S,d_\mathbb{R}}^A(\max_{a'\in A} Q_{\mathcal{M},l+1}^*)$$

$$\overset{(b)}{\leqslant} K_{d_S,d_\mathbb{R}}^A(r) + K_{d_S,W}^A(T)K_{d_S,d_\mathbb{R}}^A(Q_{\mathcal{M},l+1}^*)$$

$$\leqslant K_{d_S,d_\mathbb{R}}^A(r) + K_{d_S,W}^A(T)[K_{d_S,d_\mathbb{R}}^A(r) + K_{d_S,W}^A(T)K_{d_S,d_\mathbb{R}}^A(Q_{\mathcal{M},l+2}^*)]$$

$$\leqslant K_{d_S,d_\mathbb{R}}^A(r) + \sum_{t=1}^{H-l-2}(K_{d_S,W}^A(T))^t K_{d_S,d_\mathbb{R}}^A(r) + K_{d_S,W}^A(T)^{H-l-1}K_{d_S,d_\mathbb{R}}^A(Q_{\mathcal{M},H-1}^*)$$

$$= \sum_{t=0}^{H-l-1}(K_{d_S,W}^A(T))^t K_{d_S,d_\mathbb{R}}^A(r)$$

$$\leqslant \sum_{t=0}^{H-l-1}(K_{W,W}^A(T))^t K_{d_S,d_\mathbb{R}}^A(r)$$

where (a) follows Lemma 2 and (b) is due to the fact that the max operator is 1-Lipschitz, that is, $K_{\|\|_\infty,d_\mathbb{R}}(\max(x)) = 1$ [3]. From the definition of $r(s,a)$, we further have

$$|r(s_1,a) - r(s_2,a)| \leqslant \frac{1}{K}\sum_{k=1}^K |\ell_k(s_1) - \ell_k(s_2)| + \frac{1}{K}\sum_{k=1}^K |\mathbb{E}_{s_1'\sim T(s_1,a)}[\ell_k(s_1')] - \mathbb{E}_{s_2'\sim T(s_2,a)}[\ell_k(s_2')]|$$

$$\leqslant (L + LK_{W,W}^A(T))\|s_1 - s_2\|_2$$

where $\ell_k(s) := \mathbb{E}_{z_k\sim\widehat{P}_k}[\ell(s,z_k)]$. The first term of the second inequality comes from the Lipschitz continuity of the loss function $\ell$, which gives $|\ell_k(s_1) - \ell_k(s_2)| \leqslant L\|s_1 - s_2\|_2$ for any $k$, and the second term follows from Lemma 2 by letting $f(s) = \ell_k(s)$, which gives $K_{d_S,d_\mathbb{R}}^A(\mathbb{E}_{s'\sim T}[\ell_k(s')]) \leqslant LK_{W,W}^A(T)$ for all $k$. Since the above inequality holds for any $a \in A$, $r(s,a)$ is uniformly Lipschitz continuous in action space $A$ with a Lipschitz constant $K_{d_S,d_\mathbb{R}}^A(r) \leqslant L + LK_{W,W}^A(T)$. Thus, $K_{d_S,d_\mathbb{R}}^A(Q_{\mathcal{M},l}^*) \leqslant \sum_{t=0}^{H-l}(K_{W,W}^A(T))^t(L + LK_{W,W}^A(T))$. Since the optimal value function $V_{\mathcal{M},l}^*(s) = \max_{a\in A} Q_{\mathcal{M},l}^*(s,a)$ and the max operator is 1-Lipschitz [3], we have $K_{d_S,d_\mathbb{R}}(V_{\mathcal{M},l}^*) \leqslant K_{d_S,d_\mathbb{R}}^A(Q_{\mathcal{M},l}^*) \leqslant \sum_{t=0}^{H-l-1}(K_{W,W}^A(T))^t(L + LK_{W,W}^A(T))$. We obtain the desired result by applying Lemma 1.

$\square$

The lemma immediately implies that $V_{\mathcal{M},l}^*(\cdot)$ is $L_v$-Lipschitz for any $l$ where $L_v \leqslant \sum_{t=0}^{H-1}(K_F)^t(L + LK_F)$.

### D.4   Wasserstein distance between transitions

In this section, we bound the 1-Wasserstein distance of transition functions. Recall that the true transition dynamics $T(\cdot|s,a)$ depends on the joint distribution $\prod_{k=1}^K \widehat{P}_k$, while $T'(\cdot|s,a)$ depends on $\widetilde{P}^{K-M}\prod_{k=K-M+1}^K \widehat{P}_k$. We have the following lemma.

**Lemma 6.** *Assume Assumptions [1]-[3] hold. For any state-action pair $(s, a)$, the 1-Wasserstein distance between transition dynamics $T(\cdot|s, a)$ and $T'(\cdot|s, a)$ generated from the real FL environment and the estimated environment, respectively, is bounded by $\eta L_z \epsilon \delta$, that is,*

$$W(T(\cdot|s, a), T'(\cdot|s, a)) \leqslant \eta L_z \epsilon \delta$$

*Proof.* Let $z_1 = \{z_{1k}\}_{k=1,\ldots,K-M}$ and $z_2 = \{z_{2k}\}_{k=1,\ldots,K-M}$ denote two data sets of normal workers sampled from $\prod_{k=1}^{K-M} \widehat{P}_k$ and $\widetilde{P}^{K-M}$ respectively. Let $j = \prod_{k=1}^{K-M} j_k$ denote an arbitrary coupling between the two joint distributions that is independent across workers where $j_k$ denotes a coupling between $\widehat{P}_k$ and $\widetilde{P}$. Let $\mathcal{J}$ denote the set of all such couplings. Let $\mathcal{J}_s$ denote the collection of couplings between $T(\cdot|s, a)$ and $T'(\cdot|s, a)$ generated from the couplings of joint distributions in $\mathcal{J}$. To simplify the notation, let $s(z) := f_z(s, a)$ denote the successive state given the current state-action pair $(s, a)$ and the sampled data $z$ of normal workers. From the definition of 1-Wasserstein distance, we have

$$
\begin{aligned}
W(T(\cdot|s, a), T'(\cdot|s, a)) &\overset{(a)}{\leqslant} \inf_{j_s \in \mathcal{J}_s} \sum_{(s_1', s_2')} \|s_1' - s_2'\|_2 j_s(s_1', s_2') \\
&\overset{(b)}{\leqslant} \inf_{j \in \mathcal{J}} \sum_{(z_1, z_2)} \|s(z_1) - s(z_2)\|_2 j(z_1, z_2) \\
&= \inf_{j \in \mathcal{J}} \sum_{(z_1, z_2)} \Big\| s - \frac{1}{K}\Big( \sum_{k=1}^{K-M} \nabla_s \ell(s, z_{1k}) + a \Big) \\
&\qquad\qquad - \Big[ s - \frac{1}{K}\Big( \sum_{k=1}^{K-M} \nabla_s \ell(s, z_{2k}) + a \Big) \Big] \Big\|_2 \prod_{k=1}^{K-M} j_k(z_{1k}, z_{2k}) \\
&= \inf_{j \in \mathcal{J}} \sum_{(z_1, z_2)} \Big\| \frac{1}{K} \sum_{k=1}^{K-M} \nabla_s \ell(s, z_{1k}) - \frac{1}{K} \sum_{k=1}^{K-M} \nabla_s \ell(s, z_{2k}) \Big\|_2 \prod_{k=1}^{K-M} j_k(z_{1k}, z_{2k}) \\
&\overset{(c)}{\leqslant} \frac{\eta L_z}{K} \inf_{j \in \mathcal{J}} \sum_{(z_1, z_2)} \sum_{k=1}^{K-M} \|z_{1k} - z_{2k}\|_2 \prod_{k=1}^{K-M} j_k(z_{1k}, z_{2k}) \\
&\overset{(d)}{\leqslant} \frac{\eta L_z}{K} \inf_{j \in \mathcal{J}} \sum_{(z_1, z_2)} \sum_{k=1}^{K-M} \|z_{1k} - z_{2k}\|_2 j_k(z_{1k}, z_{2k}) \\
&\leqslant \frac{\eta L_z}{K} \sum_{k=1}^{K-M} \inf_{j_k} \sum_{(z_{1k}, z_{2k})} \|z_{1k} - z_{2k}\|_2 j_k(z_{1k}, z_{2k}) \\
&= \frac{\eta L_z}{K} \sum_{k=1}^{K-M} W(\widehat{P}_k, \widetilde{P}) \\
&\overset{(e)}{\leqslant} \frac{\eta L_z}{K}(K - M)\delta
\end{aligned}
$$

where (a) is due to the fact that we consider a restrictive collection of couplings, (b) is due to the fact that $\mathcal{J}_s$ is generated from $\mathcal{J}$, (c) follows from the smoothness of $\ell(s, z)$ with respect to $z$, (d) is due to $j_k(z_{1k}, z_{2k}) \leqslant 1, \forall k$, and (e) follows from Assumption **??**. $\qquad\square$

### D.5  Difference between expected returns

Combining the results from the previous three sections, we have the following main result.

**Theorem 1.** *Assume Assumptions [1]-[3] hold. Let $\mathcal{J}_\mathcal{M}(\pi) := \mathbb{E}_{\pi, T, \mu_0}[\sum_{t=0}^{H-1} r(s^t, a^t, s^{t+1})]$ denote the expected return over $H$ attack steps under MDP $\mathcal{M}$, policy $\pi$ and initial state distribution $\mu_0$. Let $\pi^*$ and $\widetilde{\pi}^*$ be optimal policies for $\mathcal{M}$ and $\widetilde{\mathcal{M}}$ respectively, with the same initial state distribution $\mu_0$. Then,*

$$|\mathcal{J}_\mathcal{M}(\pi^*) - \mathcal{J}_\mathcal{M}(\widetilde{\pi}^*)| \leqslant 2H\epsilon\delta[(L + L_v)\eta L_z + 2L]$$

*where $L_v \leqslant \sum_{t=0}^{H-1}(K_F)^t(L + LK_F)$ and $K_F \leqslant \epsilon \max\{|1 - \eta\alpha|, |1 - \eta\beta|\}$.*

*Proof.* By Lemma 3, $|\mathcal{J}_\mathcal{M}(\pi^*) - \mathcal{J}_\mathcal{M}(\widetilde{\pi}^*)| \leqslant 2H[(L + L_v)W(T, T') + 2L\epsilon\delta]$. From Lemma 6, we have $W(T, T') \leqslant \eta L_z \epsilon\delta$. Thus, $|\mathcal{J}_\mathcal{M}(\pi^*) - \mathcal{J}_\mathcal{M}(\widetilde{\pi}^*)| \leqslant 2H[(L + L_v)\eta L_z \epsilon\delta + 2L\epsilon\delta]$. By Lemma 5 and the comment below it, $L_v \leqslant \sum_{t=0}^{H-1}(K_F)^t(L + LK_F)$ where $K_F \leqslant \epsilon \max\{|1 - \eta\alpha|, |1 - \eta\beta|\}$. $\square$

## E   Experiments

### E.1   Experiment setup

**Datasets.**   We consider four real world datasets: MNIST [19], Fashion-MNIST [32], Balanced EMNIST [9], and CIFAR-10 [18], and a synthetic dataset. Both MNIST and Fashion-MNIST include $60,000$ training examples and $10,000$ testing examples, where each example is a $28 \times 28$ grayscale image, associated with a label from 10 classes. Balanced EMNIST includes $112,800$ training examples and $18,800$ testing examples, where each example is a $28 \times 28$ grayscale image, associated with a label from 47 classes. CIFAR-10 consists of $60,000$ color images in 10 classes of which there are $50,000$ training examples and 10,000 testing examples. Details about the synthetic data are given in Appendix E.2. For the *i.i.d.* setting, we randomly split the dataset into $K$ groups, each of which consists of the same number of training samples. For the *non-i.i.d.* setting, we follow the method of [11] to quantify the heterogeneity of local data distribution across clients. Suppose there are $C$ classes in the dataset, e.g., $C = 10$ for the MNIST, Fashion-MNIST, and CIFAR-10 datasets. We evenly split the worker devices into $C$ groups, where each group is assigned $1/C$ of training samples as follows. A training instance with label $c$ is assigned to the $c$-th group with probability $q \geqslant 1/C$ and to every other group with probability $(1 - q)/(C - 1)$. Within each group, instances are evenly distributed. A higher $q$ indicates a higher *non-i.i.d.* degree. We set $q = 0.5$ as the default *non-i.i.d.* degree. To demonstrate the power of distribution learning, we assume that the set of attackers share $m$ true data points sampled from the training instances assigned to them. We set $m = 200$ for MNIST and Fashion-MNIST, $m = 500$ for EMNIST, and $m \in \{500, 5000\}$ for CIFAR-10.

**Federated learning setting.**   We adopt the following parameters for the federated learning models: learning rate $\eta = 0.01$ (0.05 for EMNIST and the synthetic data), total number of workers $= 100$, number of attackers $= 20$, subsampling rate $= 10\%$, and number of total epochs $= 1000$. For the MNIST, Fashin-MNIST, and EMNIST datasets, we train a neural network classifier consisting of 8×8, 6×6, and 5×5 convolutional filter layers with ReLU activations followed by a fully connected layer and softmax output. The cross-entropy loss is used to optimize the model. For CIFAR-10, we use the ResNet-18 model [15]. We set the local batch size $B = 128$. We implement the FL model with PyTorch [20] and run all the experiments on the same 2.30GHz Linux machine with 16GB NVIDIA Tesla P100 GPU. We simulate subsampling and local data sampling with different random seeds in each test run. Error bars are reported in Figure **??**(c) in the main paper. We set cross-entropy as our default loss function, and stochastic gradient descent (SGD) as our default optimizer.

**Baselines.**   We compare our RL-based attack (RL) with no attack (NA), and the state-of-the-art model poisoning FL attack methods: explicit boosting (EB) [5], inner product manipulation (IPM) [33], and local model poisoning attack (LMP) [11]. The EB attack [5] is originally proposed for the targeted setting. We adapt it to the untargeted setting by using empirical loss as the objective, which is optimized through multi-step gradient ascent using attackers' local data, where the number of steps is 5 and the step size equals to the FL learning rate $\eta$. The model update is then boosted by a factor of $\frac{K}{M}$. We compare our RL-based attack with the full knowledge LMP [11], where the attackers have access to not only the aggregation rule but also all normal workers' updates. We use the LMP attack tailored to Krum when the Krum defense is used, and the LMP attack tailored to coordinate-wise median when the clipping median defense or the geometric median defense is used. Further, we implement the adaptive version of LMP introduced in [8], which requires the attackers to know the server's updates derived from its root data, as a baseline against the FLTrust defense [8]. In our implementation of IPM [33], we set the default boosting factor (i.e., $\epsilon$ in [33]) as 5.

We consider four representative robust aggregation rules of different types [25]: Krum [6] and geometric median [21], both of which apply client-wise filterings to model updates, coordinate-wise median [34], which adopts a dimension-wise filtering, and FLTrust [8], which requires the server to

collect a small training dataset $D_0$ (called root dataset). In the experiments, we actually consider an extension of the vanilla coordinate-wise median where a norm clipping step [28] is first applied. This gives a more powerful defense as we observed in experiments. We set the default clipping threshold to 2. In geometric median [21], we set the iteration number of the smoothed Weiszfeld algorithm for computing the geometric median [21] to 10 to balance effectiveness and efficiency. In FLTrust, the root data is used to calculate a server model update $g_0 = \frac{1}{|D_0|} \sum_{z \in D_0} [\nabla_\theta \ell(\theta; z)]$ in each epoch. The aggregation weight of each received client' update is then determined through its ReLU-clipped cosine similarity with $g_0$. Given that the server has no access to the true training data distribution, the root dataset is often biased in practice. We adopt the approach in [8] to model such bias. Among the $|D_0|$ root data samples, a fraction $q_0$ of them are sampled from a certain class $c$ in the training data, and the rest are sampled from other classes with equal probabilities. For a dataset with $C$ classes, $D_0$ is unbiased only when $q_0 = 1/C$. We set the size of root dataset $|D_0| = 100$ following [8].

**Distribution learning setting.**   In distribution learning, we set the step size for inverting gradients $\eta' = 0.05$, the total variation parameter $\beta = 0.02$, optimizer as Adam, the number of iterations for inverting gradients $max\_iter = 10,000$, and learn the data distribution from scratch. The number of steps for distribution learning is set to $\tau_E = 100$. 32 images are reconstructed (i.e., $B' = 32$) and denoised in each FL epoch. If no attacker is selected in the current epoch, the aggregate gradient estimated from previous model updates is reused for reconstructing data. To build the denoising autoencoder, a Gaussian noise sampled from $0.3\mathcal{N}(0, 1)$ is added to each dimension of images in $D_{reconstructed}$, which are then clipped to the range of [0,1] in each dimension.

**Policy learning setting.**   In policy learning, we implement our simulated environment with OpenAI Gym [7] and adopt OpenAI Stable Baseline3 [23] to implement Twin Delayed DDPG (TD3) [12] and Proximal Policy Optimization (PPO) [24] algorithms. We find that TD3 gives better results in most cases and report the results for TD3 below. The default parameters are described as follows: the length of simulating environment = $1,000$ epochs, policy learning rate = $1e-7$, the policy model is $MultiInputPolicy$, batch size = 256 and gamma = 1 for updating the target networks.

As described in Section 4.3, we compress the MDP state to include the parameters of the last hidden layer of $\theta^{t(\tau)}$ and the number of attackers sampled, $m^{t(\tau)}$, where each last hidden layer parameter is in $[-\infty, +\infty]$ and $m^{t(\tau)}$ is in $\{0, \ldots, 10\}$. In our experiment, we restrict all attackers to take the same action in each epoch. In solving the local search problem, we fix the number of trajectories $G = 1$ and the size of minibatch $\widetilde{B} = 200$ (except for FLTrust where $\widetilde{B} = 500$).

For the Krum, clipping median, and geometric median defenses, the local search objective is $F(\theta) = \mathbb{E}_{z \sim \widetilde{P}}[\ell(\theta; z)]$ (i.e., $\lambda = 0$). In this case, the action space becomes $(\gamma, E)$, where $\gamma \in [0, 10]$ and $E \in \{0, \ldots, 20\}$ for the Krum defense, and $\gamma \in [0, 10]$ and $E \in \{0, \ldots, 50\}$ for the clipping median and geometric median defenses. Since TD3 can only be applied to a continuous action space, we consider a continuous interval for $E$ (e.g., $E \in [0, 20]$ for Krum) when updating the policy and round its value to an integer in the feasible range before the action is applied.

For FLTrust, we consider two cases, when the attackers have access to the server's root data $D_0$ or equivalently, the model update $g_0$ in each epoch, and when they only know how $D_0$ is sampled from the true training data distribution. Note that even the former setting is more realistic than the adaptive LMP setting in [8], which also requires access to normal workers' updates. In the former case, we slightly modify the local search method described in Section **??** by fixing $\gamma(\theta^{t(\tau)}) = \|g_0(\theta^{t(\tau)})\|_2$ and considering the same local search objective $L(\theta) := (1-\lambda)F(\theta) + \lambda \cos(\theta^{t(\tau)} - \theta, g_0(\theta^{t(\tau)}))$ with the extra constraint that $\|\theta^{t(\tau)} - \theta\|_2 \leqslant \|g_0(\theta^{t(\tau)})\|_2$. This is because FLTrust normalizes all the local model updates using the magnitude of the root update. In the latter case, we use the same objective but approximate $g_0(\theta^{t(\tau)})$ with $\mathbb{E}_{z \sim q_0 \widetilde{P}}[\nabla_\theta \ell(\theta^{t(\tau)}; z)]$, where $q_0$ models the bias of root data, which is assumed to be known to the attackers. In both cases, the action space is then $(E, \lambda)$ with $E \in \{0, \ldots, 20\}$ and $\lambda \in [0, 1]$. We further find that when the root data $D_0$ is known (or can be well approximated), the RL-based attack can be made more efficient by considering an alternate local search objective $L(\theta) := (1-\lambda)F(\theta) - \lambda F_0(\theta)$, where $F_0(\theta) = \frac{1}{|D_0|} \sum_{z \in D_0} [\ell(\theta; z)]$ is the empirical loss associated with the root data. Intuitively, the attackers aim to push the model parameters towards the region that can overfit the root data.

In our experiments, the initial model for all training episodes is set as the first model the attackers received from the actual FL environment. We assume that the server waits for 72 seconds to receive

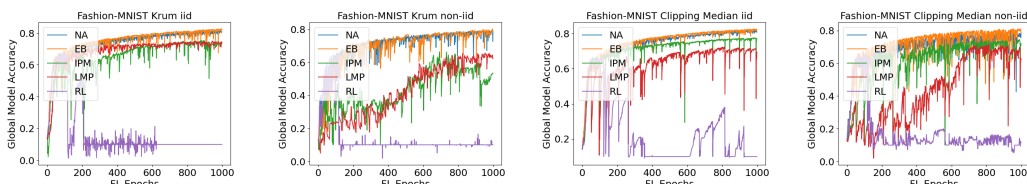

Figure 1: A comparison of global model accuracy on Fashion-MNIST under Krum and clipping median for both *i.i.d.* data and *non-i.i.d.* data. All parameters are set as default.

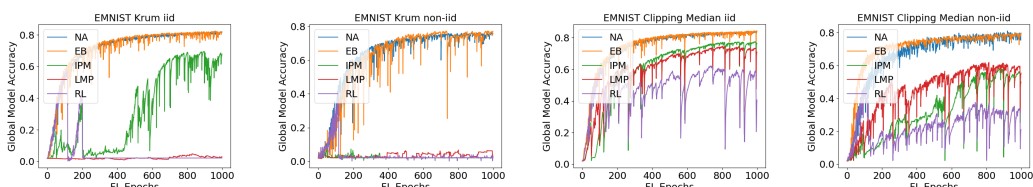

Figure 2: A comparison of global model accuracy on EMNIST under Krum and clipping median for both *i.i.d.* data and *non-i.i.d.* data. All parameters are set as default.

the updates from the workers before performing a model aggregation, which allows $80,000$ total time steps (i.e., 80 episodes) of policy learning for Krum, $40,000$ total time steps (i.e., 40 episodes) of policy learning for clipping median, and $40,000$ total time steps (i.e., 40 episodes) of policy learning for FLTrust within 400 FL epochs in our experiment setting. It is more time consuming to train an RL policy for clipping median and FLTrust because large attack bounds need to be considered. See E.2 for a detailed comparison of the running time of different stages of the RL-based attack under different defense scenarios.

**Attack execution setting.** Both the distribution learning and policy learning phases in the RL-based attack start at the first FL epoch. The former ends at the 100th FL epoch when RL-based attack starts. All other attacks start at epoch 0. For fair comparisons, we fix all the random seeds for generating the initial model and the root data (for FLTrust), subsampling, and local data sampling when evaluating different attacks. We observe that both EB and RL can occasionally produce NaNs in model updates, which when incorporated by the server, can lead to bad models in all future steps. This produces unrealistic attack scenarios as NaNs can be easily detected by the server. To have a fair comparison with other attacks, we use the built-in VecCheckNan Wrapper in OpenAI Stable Baseline3 [23] to detect abnormal values. We assume that attackers take less ambitious actions (i.e., $(0.5\gamma, E-1)$) in that epoch once they detect a NaN value. If $E = 0$ or $\gamma = 0$, the attackers send $\widetilde{g}^{t(\tau)} = \mathbf{0}$ to the server.

### E.2 More experiment results

**Attack performance on other datasets.** Figures 1 and 2 compare the test accuracy of the global model under different attacks when the server uses Krum or clipping median as the defense for the Fashion-MNIST and EMNIST datasets. We consider both *i.i.d.* and *non-i.i.d.* ($q = 0.5$) settings. Our RL-based attack constantly outperforms other baselines by a large margin in all the settings. We observe that in most cases, all attacks are more effective in the *non-i.i.d.* setting. This is mainly because a higher degree of local data heterogeneity increases the variance across normal workers' updates, making it more difficult to filter out adversarial updates. Further, clipping median, which adopts both dimension-wise filtering and client-wise norm clipping to model updates, provides a stronger level of defense than Krum, which only applies client-wise filtering to model updates. In particular, our attack can reduce the model accuracy to an extremely low level under the Krum defense, depending on the number of classes of the dataset used ($\sim 10\%$ for Fashion-MNIST and $\sim 2\%$ for EMNIST).

Figure 3 compares the test accuracy of the global model under different attacks for the CIFAR-10 dataset in the *i.i.d.* setting. Here we assume that our RL-based attack does not perform distribution learning, and the attackers use their local data to train the attack policy and start to execute attack at epoch 100. This is mainly due to the fact that image reconstruction for CIFAR-10 takes prohibitive

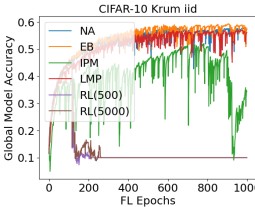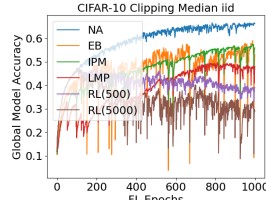

Figure 3: A comparison of global model accuracy on CIFAR-10 under the Krum and clipping median defenses. The RL policy is trained using 500 or 5,000 local samples without distribution learning.

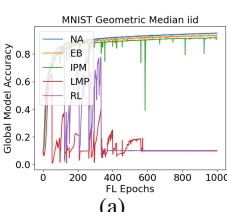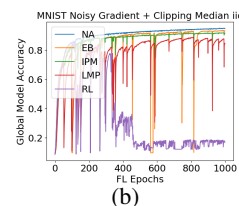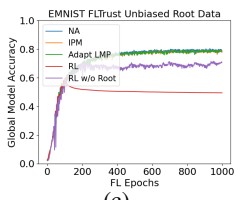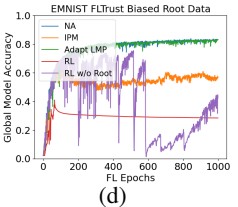

Figure 4: More results on different defenses. (a) Attack performance on MNIST under the geometric median defense. (b) Attack performance on MNIST under the clipping median defense and noisy gradients. (c) and (d) Attack performance on EMNIST under FLTrust defense with unbiased and biased root data.

amount of time in our experiment environment. Further, state-of-the-art gradient inversion attacks either cannot reconstruct a large batch of images for CIFAR-10 accurately or have not made their code available yet. We consider two cases where 500 and 5,000 local samples are used to train the attack policy, respectively. We observe that in both cases, our approach surpasses all the baselines. In particular, the RL policy trained using only 500 local samples quickly drives the model accuracy to a very low level ($\sim\%9.52$) under the Krum defense.

**Attack performance under geometric median.** We compare the attack performance of RL-based attack and other baselines (i.e., NA, EB, IPM, and LMP) against geometric median [21] on MNIST dataset in the *i.i.d.* setting. As shown in Figure 4(a), RL-based attack and LMP significantly outperform other baselines. Further, although our RL-based attack starts attacking at the 100th epoch, it quickly drives the model accuracy to a very low level, while LMP takes much longer time to achieve similar attack performance.

**Attack performance under noisy gradients.** We also compare the attack performance of our RL-based attack and other baselines against clipping median aggregation (with the clipping threshold set to 2) under noisy gradients [31]. In particular, the server injects noise into the global model parameters shared with clients, where the noise is sampled from a Laplace distribution [2] (i.e., double exponential distribution) with 0 mean and $1e-4$ exponential decay. We observe that although adding noise indeed decreases the quality of reconstructed images, distribution learning is still effective for the MNIST dataset. Further, our RL-based method still outperforms other baselines in this setting as shown in Figure 4(b).

**Attack performance under FLTrust.** We compare the attack performance of our RL-based attack with and without access to server's root data (details are given in E.1 policy learning setting) and other baselines (i.e., NA, IPM, and adaptive LMP) against the FLTrust defense on the EMNIST dataset. For RL-based attacks, the attackers use 5,000 local data samples to simulate the environment and skip the distribution learning phase, and start attacking at FL epoch 100. All the baselines start from the beginning of FL. We consider both the cases when the root data are unbiased ($q_0 = 1/47$) and when they are biased against a single class ($q_0 = 0.3$). In the former case, our attack with access to root data leads to a significantly low test accuracy ($\sim50\%$) as shown in Figure 4(c), while other attacks, including RL-based attack without access to root data, have limited effect against FLTrust. This is due to the fact that when the root data are unbiased and representative of the true training dataset, the root update $g_0$ in each epoch provides a good estimate of the right direction for model updates, making it difficult to reverse the trend. On the other hand, when the root data is biased, which is

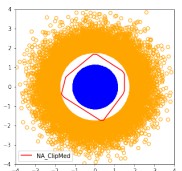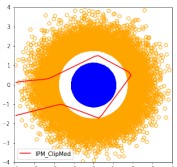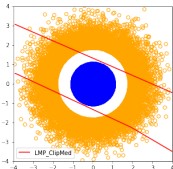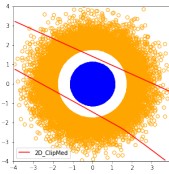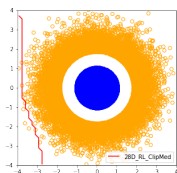

Figure 5: Classification boundaries of the final model on the synthetic data under various attacks and the clipping median defense. The classification accuracy of the final model: 100% (NA), 96.70% (IPM), 89.04% (LMP), 88.04% (RL with 2d actions), and 68.90% (RL with 28-dimensional actions). All parameters are set as default.

likely to happen in practice, the root updates are less representative or even misleading. As shown in Figure 4(d), our RL-based attack with root data access becomes more effective as expected. Our RL-based attack without root data also achieves significant although unstable attack performance. Here we ignore the second term in the local search objective $L(\theta)$ by fixing $\lambda = 0$ to minimize the impact of inaccurate estimate of $g_0$.

**Actual runtime comparison.** The actual runtime varies across the FL environment, the training method used, and most importantly, the amount of computational resource available. The tables below report the numbers from our current experiment settings (see Appendix E.1) and the way the simulator is implemented (clients are simulated sequentially in each FL epoch).

For MNIST, Fashion-MNIST, and EMNIST, distribution learning takes around 100 seconds to reconstruct a batch of 32 images and we construct 50 batches within 2 hours. Note that multiple batches can be generated from a single gradient. We start policy training from the beginning of FL training, and we set 8 hours limit for policy training. It takes around 0.05 seconds to simulate a single FL epoch with 10 sampled clients without parallelization. Total training steps vary across defense policies as stated in Appendix E.1.

With the above numbers, if we assume that each FL epoch takes 72 seconds to finish and there are in total of 1000 FL epochs during FL training, then distribution learning will end before the 100th FL epoch and policy training ends by the 400th FL epochs, and the total FL training time is around 20 hours.

| Stages | FL Epochs | Real Time |
|---|---|---|
| Distribution Learning | 100 | $\leqslant 2$ hours |
| Policy Learning | 400 | $\leqslant 8$ hours |
| Total FL Training | 1000 | 20 hours |

Table 1: The running time of each stage in our RL-based attack in terms of FL epochs and real running time for small networks.

For CIFAR-10, we do not perform distribution learning in this work and policy learning alone takes about 20 hours in our experiment environment as we use a much bigger network (i.e., Resnet-18). However, we expect that once equipped with more powerful devices, the training time can be significantly reduced by parallelly simulating multiple clients using multiprocessing and multiple episodes using vectorized environments, which will make it possible to simulate large FL systems.

In terms of attack executing time, for MNIST with clipping median defense, IPM takes around 0.25 seconds to execute an attack in each FL epoch, LMP takes around 7.7 seconds, EB takes around 0.5 seconds. For CIFAR-10 with clipping median defense, IPM takes around 2.55 seconds to execute an attack in each FL epoch, LMP takes around 30 seconds, EB takes around 5.5 seconds. The execution time of our RL-based method varies over the action space used and it takes around 5.8 seconds and 6 seconds for MNIST and CIFAR-10 respectively with the default action space described in Section **??**. Given that each FL epoch typically lasts a minute or longer (72 seconds in our experiment), a few seconds of search time is completely acceptable. We observe that for defenses such as Krum, it suffices to use the gradients of the last two layers of model parameters as the action. This approach does not require any online searching and reduces the attack execution time to 0.5s.

| Attacks | MNIST | CIFAR-10 |
|---------|-------|----------|
| IPM | 0.25s | 2.5s |
| LMP | 7.7s | 30s |
| EB | 0.5s | 5.5s |
| RL | 5.8s | 6s |

Table 2: Execution time of various attacks against the clipping median defense for the MNIST and CIFAR-10 datasets.

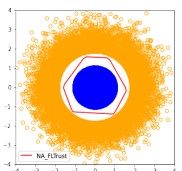 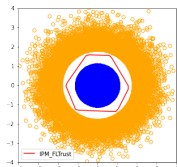 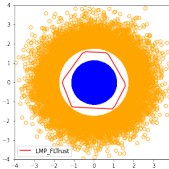 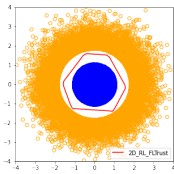 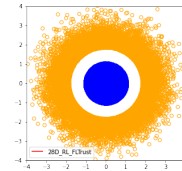

Figure 6: Classification boundaries of the final model on the synthetic data under various attacks and the FLTrust defense. The classification accuracy of the final model: 100% (NA), 100% (IPM), 100% (LMP), 100% (RL with 2d actions), and 68.90% (RL with 28-dimensional actions). All parameters are set as default.

| Attacks | 5% attackers | 10% attackers | 20% attackers |
|---------|--------------|---------------|---------------|
| NA | 99.70% | 99.02% | 99.86% |
| IPM | 99.66% | 88.88% | 68.96% |
| EB | 99.68% | 84.26% | 70.06% |
| LMP | 99.68% | 89.38% | 69.04% |
| RL | 68.90% | 68.90% | 68.90% |

Table 3: Global model accuracy under various attacks and the Krum defense on the synthetic dataset.

**Results for the synthetic data.** In addition to the four real datasets discussed above, we also consider a two-dimensional synthetic dataset and a small network with 28 model parameters to demonstrate the full potential of our RL-based attack framework (i.e., without state and action compression). We generate the synthetic data based on the method described in [26]. In particular, we generate $55,000$ data instances (including $50,000$ training instances and $5,000$ testing instances), where for each instance $z = (x, y)$, the data $x \in \mathbb{R}^2 \sim \mathcal{N}(\mathbf{0}, I)$ and its label $y = \text{sign}(\|x\|_2) - 2$. Each worker has $500$ data instances. We train a multilayer perceptron (MLP) with two hidden layers of size four and two, respectively, and use ReLU as the activation function. For our RL-based attack, we consider both the 2-dimensional action space $(\gamma, E)$ discussed above as well as the general 28 dimensional action space where the attackers directly decide $\widetilde{g}^{t(\tau)}$ to be sent to the server in each epoch. In both cases, the state space includes the full 28 model parameters and the number of attackers in each epoch. Policy learning takes $8,000$ total time steps (i.e., 8 episodes) to learn the policy, within 10 FL epochs. The attackers use their local data ($10,000$ samples) to build a simulated environment without using distribution learning, and start attacking at epoch 0. We fix all random seeds for a fair comparison across different attacks.

Figure 5 and Figure 6 illustrate the classification boundaries at the end of a federated learning episode for all the attacks when the clipping median defense and the FLTrust defense are applied respectively. The root dataset $D_0$ for FLTrust is assumed to be known for RL-based attacks. We observe that all the baseline methods and our RL-based attack with 2d actions have limited effect under clipping median and completely fail under FLTrust. On the other hand, the RL-based attack with the full 28-dimensional action space reduces the classification accuracy to $68.90\%$ (worst-case accuracy for the given environment) under both defenses. These results indicate the potential of considering large state and action spaces in our RL-based attack when equipped with more computational power and longer training time.

Table 3 shows how the global model accuracy under different attacks and the Krum defense varies over the number of attackers. The results show that our approach is effective even when the fraction of malicious clients is as low as 5%.