# OpenReview forum: "Learning to Attack Federated Learning: A Model-based Reinforcement Learning Attack Framework"
_NeurIPS.cc/2022/Conference — NeurIPS 2022 Accept_

### Official Review · Reviewer_kyEC · 2022-07-08

**Rating:** 6
**Confidence:** 3
**Soundness:** 2 fair
**Presentation:** 2 fair
**Contribution:** 2 fair

**Summary:**

This paper proposes to simulate a RL environment to learn a poisoned gradient, which will be sent to the server for aggregation by some attackers. This paper achieves SOTA attack performance on MNIST and Fashion-MNIST.

**Questions:**

-	In figure 3, the RL curve converges to random guess and has no oscillation, which is very weird to the reviewer. It seems like a NaN in gradient.
-	In figure 4 c), the RL histogram performs very weird, especially 0.3, RL here seems to have no std? This is nearly impossible for RL.
-	In experiments settings, there are in total 100 works with 20 attackers in it, this setting is so weird.


**Strengths And Weaknesses:**

Major weakness:
-	The novelty is limited. Why do we need RL here is not clear, compared with some adversarial manipulation on the gradients, e.g., directly maximizing the “reward” function w.r.t. gradients. RL is unstable and hard to train, the necessity should be addressed, otherwise, it seems like simply an adaptation of some techniques.
-	The evaluation is far from enough. This paper should also include some results on colored images, e.g. CIFAR10.
-	 In practice, noisy gradients may also be considered to provide better privacy. This paper proposes a technique heavily rely on the accuracy of gradients, so, an ablation study is suggested.
-	This paper proposes a technique that seems to require a large amount of malicious users. An ablation study on the number of malicious users is suggested.
-	Given the limited contribution of this paper, it is suggested that this paper provides some investigation on why gradients generated by RL can be better, what kind of properties it possesses, what is its advantages over other methods, or under what settings this method has an overwhelming advantage. This paper also addresses some potential future works, many of them can be integrated organically in this paper to increase this paper’s contribution, e.g backdoor attacks.
-	Achieve SOTA on this task on some non-real world dataset is less significant, this paper is suggested to dig out some insights from other aspects, instead of just a SOTA.

Minor weakness:
- Typos & Grammars: Line 128; Line 304;
- Writing: The reviewer doesn’t see any insights w.r.t data heterogeneity in section 4, while that is in the section title.
- It seems that selected malicious clients are required to perform much much much more computations compared with other clients. Although this paper includes a paragraph discussing (line 266- line 274). Some quantitative measure may be preferred, e.g. seconds for FL epoch, seconds for RL epoch.

---

> ### Author Response · Authors · 2022-08-02
> **Response to Reviewer kyEC**
>
> Thanks for your valuable and constructive feedback and comments. We will address your concerns point by point in the following.
>
> Q1: Why use RL? Why is our method better than myopic methods?
>
> A1: In this work, we consider online model-poisoning attacks against federated learning, which is a sequential decision making problem under uncertainty (from the attacker’s perspective) and RL is a natural framework for it. Compared with previous one-shot methods (e.g., LMP, IPM, and EB), the goal of an RL attack is non-myopic, that is, maximizing the attackers’ long-term expected rewards. A one-step optimizing goal is usually sub-optimal, especially when a robust aggregation rule is adopted.  We observed in our experiments that the FL system can often recover quickly from a bad model under myopic attacks while RL can slow down the process (see Figures 3(c) and 3(d) in the paper). With potential strong defenses, it is crucial to attack in a “low-and-slow” way so that the attack effects will accumulate even if the one step attack ability is limited by the defense. In an FL system, since the next global model depends on the current one and the clients’ actions, it is natural to model it as a Markov decision process, which captures the evolution of the global model during FL training.
>
> We choose deep-RL with dimension reduction (see Section 3.3 and Appendix D.1) to solve the attacker’s MDP, since it is typically more efficient than traditional dynamic programming and linear programming based methods, at the cost of being sample-inefficient and unstable as the reviewer points out. To solve the first problem, we have considered a model-based approach by building a simulator using the learned data distribution. To solve the second problem, we can set up a separate testing environment to identify the best trained policies as we briefly mentioned in the experiment section. On the other hand, we observed in our experiments that a sufficiently trained RL policy can typically obtain strong attack performance despite the instability.
>
> Q2: Why RL attack achieves a flat line.
>
> A2: In this case, our RL attack drives the model accuracy to a very low level (~10%) due to the loss being extremely large. However, it is not due to a NaN in the gradient vector, as we adopt a NaN detection technique in our experiments, and the attacker will take a less ambitious action if a NaN is encountered (see Appendix D.1 Attack execution setting). As we observed in the experiments, the RL attack can quickly lead the server to a ‘bad’ model, while each gradient it sends is still legal. This again shows the advantage of the RL attack over myopic attacks, i.e., finding a shortest path (multiple steps into the future) towards a target model instead of finding a one-step gradient (after aggregation) that points to a bad model.
>
>
> Q3: Why the variance is low for RL.
>
> A3: We observe that when the loss of the global model is beyond a certain value, model accuracy will be constant or close to a low point. Thus, when our attack dramatically damages the FL training, the final accuracy will be similar and low. This explains why the variance of RL results is low or close to 0 in Figure 4 (c).
>
> Q4: Performance under different number of attackers
>
> A4: A 20% of attackers is a typical setting in traditional untargeted model poisoning attacks against FL with robust defense, where a sufficient number of attackers is typically needed to achieve a strong attack performance [1-3]. For example, the default setting is 20% in [1] and 40% in [2]. Although [3] uses 10% of attackers, they consider a targeted attack. The table below shows that our RL-based approach is still effective even with a smaller number of attackers as it can exploit the vulnerability of an FL system more effectively.
>
> MNIST dataset with Krum defense
>
>                            5% attackers         10% attackers         20% attackers
>            NA                94.24%                94.35%               93.68%
>            IPM               94.2%                 93.84%               90.44%
>            LMP               92.08%                91.06%               88.57%
>            Our Method        94.09%                9.8%                 11.35%

---

> > ### Author Response · Authors · 2022-08-02
> > **Response to Reviewer kyEC**
> >
> > To further demonstrate the power of our RL based method, we report the results on a synthetic dataset (see Appendix D.2) below. For the synthetic data, we use a small network with 28 parameters for FL training, and the attacker directly manipulates all of them (a 28-dimensional action space). The results show that our approach is effective even when the fraction of malicious clients is as low as 5%.
> >
> > Synthetic dataset with Krum defense
> >
> >                        5% attackers                 10% attackers
> >          NA              99.70%                        96.02%
> >          IPM             99.66%                        88.88%
> >          EB              99.68%                        84.26%
> >          LMP             99.68%                        89.38%
> >          Our Method      68.90%                        68.90%
> >
> > Q5: No insights w.r.t. data heterogeneity in Section 4.
> >
> > A5: Theorem 1 proved in Section 4 captures the impact of inaccurate data distribution on the attack performance, where the inaccuracy comes from both the inaccurate distribution learning as well as data heterogeneity, as the latter makes it more difficult to infer accurate global data distribution. We note that data heterogeneity introduces challenges to both attacks and defenses. In our experiments, we showed that our attack is still effective even under non-iid data distribution across clients.
> >
> > For a discussion on the CIFAR-10 results, please see our response to u8VB Q2.
> >
> > We have included results for noisy gradient, please see our response to Gb3o Q1.
> >
> > For results about running time, please see our response to u8VB Q1.
> >
> >
> > [1] Fang, Minghong, et al. "Local model poisoning attacks to {Byzantine-Robust} federated learning." USENIX Security 2020.
> >
> > [2] Xie, Cong, Oluwasanmi Koyejo, and Indranil Gupta. "Fall of empires: Breaking byzantine-tolerant sgd by inner product manipulation." UAI 2020.
> >
> > [3] Bhagoji, Arjun Nitin, et al. "Analyzing federated learning through an adversarial lens." ICML 2019.

---

> > > ### Comment · Reviewer_kyEC · 2022-08-09
> > > **Thanks for your response**
> > >
> > > I appreciate the authors' response and informative clarification. After reading the rebuttal and other reviewers' comments, most of my concerns have been addressed. I will change my score. Thank you.

---

> > > > ### Author Response · Authors · 2022-08-09
> > > > **Thank you for updating your review**
> > > >
> > > > We are glad that we have addressed most of your concerns. We appreciate the updating score and valuable feedback.

---

### Official Review · Reviewer_Gb3o · 2022-07-10

**Rating:** 5
**Confidence:** 4
**Soundness:** 3 good
**Presentation:** 3 good
**Contribution:** 3 good

**Summary:**

The works aims to propose an adaptive model poisoning attack method against federated learning systems. To this end, the paper proposes a model-based reinforcement learning attack framework. To utilize the limited global knowledge, the proposed framework first conduct the distribution learning step, which approximates the distribution of the clients’ aggregated data. Then in the policy learning step, the malicious clients get the attack policy based on the learned FL environment and reinforcement learning, and finally generate attack actions. The authors also theoretically analyze the performance loss of the attack policy due to the inaccurate distribution learning and data heterogeneity. Through experiments with real-world datasets, the authors validates that the proposed framework could successfully poison the global model even when the server adopts a robust aggregation rule.

**Questions:**

* It seems that the robust aggregation oracle proposed in [1] have better defense performance against the data/model poisoning attacks. Is there any reason not including it in the experiments?

[1] Pillutla, Krishna, Sham M. Kakade, and Zaid Harchaoui. "Robust aggregation for federated learning." arXiv preprint arXiv:1912.13445 (2019).


POST REBUTTAL COMMENTS: Part of my concerns have been addressed and the authors promised to solve the others in the future.

**Limitations:**

As mentioned in the weaknesses, the distribution learning step could lead to significant attack performance loss. The defenses against gradient leakage attacks should be considered.

**Strengths And Weaknesses:**

Strengths
* Privacy related issues in federated learning is getting more important. The paper proposes an adaptive attack to poison the global model by integrating distribution learning and RL-based policy learning.
* The paper provides theoretical analysis on the effect of inaccurate distribution learning and heterogeneous local data distributions on the performance loss of the attack policy.
* The proposed RL-based attack framework could be potentially extended to different attacks and be applied to other learning scenarios.

Weaknesses
* There have been proposed several defenses against gradient leakage attacks, e.g., adding noise, and most gradient leakage attacks are only effective on small batches. Both leads to lower estimation accuracy in the distribution learning step. Besides, the Theorem 1 shows that the performance loss is linearly related to the estimation accuracy $\delta$, which makes the distribution learning step crucial.
* Besides the robust aggregation rules, several defenses are proposed to defend against targeted/untargeted poisoning attacks based on abnormal detection (e.g., SPECTRE [1]). The experiments should consider this kind of defenses.
* More complicated datasets should be considered (e.g., CIFAR-10) in the experiments since the distribution learning could be less accurate.

[1] Hayase, Jonathan, et al. "SPECTRE: defending against backdoor attacks using robust statistics." arXiv preprint arXiv:2104.11315 (2021).

---

> ### Author Response · Authors · 2022-08-02
> **Response to Reviewer Gb3o**
>
> Thank you for your insightful and valuable comments. We will address your concerns point by point in the following.
>
> Q1: Defense against gradient leakage by adding noise.
>
> A1: As mentioned in [1] and [2], noise is typically added to the gradients from a client to the server in order to prevent privacy leakage. In our setting, an attacker infers the gradient by using two adjacent global models broadcasted by the server. Thus, the server needs to add noise to the global model to prevent the attacker from inferring the accurate gradient. We tried adding noise to the broadcasted global model and using clipping median as an aggregation rule. For MNIST, adding noise indeed decreased the quality of reconstructed images, but many of them are still recognizable and our RL based method still outperforms other baselines in this setting.
>
> MNIST + Clipping Median + Noisy Gradient
>
>                                Accuracy
>       No Attack                 94.84%
>       IPM                       91.58%
>       EB                        93.29%
>       LMP                       84.74%
>       Our Method                16.89%
>
> Q2: Abnormal detection-based defense.
>
> A2: Thanks so much for pointing out a possible next step for our work. Currently, we do not consider detection-related defenses. However, as long as the attacker has knowledge of the detection mechanism used by the server, it could still build a simulator of the FL system and derive an effective attack using our method. A new challenge is that an abnormal detection-based defense usually requires maintaining historical records for each client [4]. To attack such a defense, our policy learning method needs to be further expanded by either explicitly including history information in the state or implicitly storing it in a hidden state by utilizing a recurrent structure.
>
> Q3:Robust aggregation for federated learning.
>
> A3: As suggested by the reviewer, we added an experiment investigating the geometric median (GM) based robust aggregation proposed in [3].
>
> MNIST + Geometric Median [3] + 20% Attackers
>
>                                Final Accuracy
>       No Attack                    95.03%
>       IPM                          91.62%
>       EB                           93.48%
>       LMP                          9.74%
>       Our Method                   10.1%
>
> We observe that although GM can successfully defend against IPM and EB, it is not robust under our RL-based attack.
>
> For a discussion on the CIFAR-10 results, please see our response to u8VB Q2
>
> [1] Huang, Yangsibo, et al. "Evaluating gradient inversion attacks and defenses in federated learning." NeurIPS 2021.
>
> [2] Wei, W., Liu, L., Loper, M., Chow, K. H., Gursoy, M. E., Truex, S., & Wu, Y. (2020). A framework for evaluating gradient leakage attacks in federated learning. arXiv preprint arXiv:2004.10397.
>
> [3] Pillutla, K., Kakade, S. M., & Harchaoui, Z. (2022). Robust aggregation for federated learning. IEEE Transactions on Signal Processing, 70, 1142-1154.
>
> [4] Li, S., Cheng, Y., Liu, Y., Wang, W., & Chen, T. (2019). Abnormal client behavior detection in federated learning. arXiv preprint arXiv:1910.09933.

---

> > ### Author Response · Authors · 2022-08-09
> > **Response to Reviewer Gb3o**
> >
> > We are glad that we were able to address some of your concerns.
> >
> > We would like to clarify one comment regarding inaccurate distribution learning. Since the attack performance is related to the distribution estimation accuracy as we proved in Theorem 1, a few inaccurately learned images (e.g., with noisy pixels or wrong labels) won't significantly influence our policy learning and the final attack performance. Further, as we show in Figure 4(a) in the paper and the CIFAR-10 results above (please see our response to u8VB Q2), even without distribution learning, our attack still obtains good performance using limited local data owned by attackers (200 images for MNIST and 500 images for CIFAR-10).
> >
> > We also want to thank the reviewer for sharing with us the abnormal detection paper (SPECTRE [1]). We have checked this paper and found that SPECTRE cannot be directly used in our setting. First, it targets backdoor attacks rather than model poisoning attacks we considered in this work. Second, it requires access to the whole corrupted training dataset, which does not hold in federated learning.
> >
> > [1] Hayase, Jonathan, et al. "SPECTRE: defending against backdoor attacks using robust statistics." arXiv preprint arXiv:2104.11315 (2021).

---

### Official Review · Reviewer_u8VB · 2022-07-10

**Rating:** 6
**Confidence:** 3
**Soundness:** 3 good
**Presentation:** 2 fair
**Contribution:** 2 fair

**Summary:**

- The authors propose a RL-based attack framework against federated learning (FL) systems.
- The framework learn distribution of the dataset using the data of malicious workers and conduct model-based RL using the learned distribution.
- The authors provide a theoretical analysis of the effect of inaccurate distribution learning on the difference in the optimal return.
- The authors conduct model poisoning attack to FL system of MNIST and EMNIST learning under Krum and Clipping Median scheme. The experimental results show that the proposed RL-based method outperform the baseline methods.

**Questions:**

- To apply the attack method on an online FL system, the actual runtime of the attack method  is very important. Even though they suggest a technique for scalability, many RL-based applications suffer from a large amount of computation and runtime. I think an analysis on the actual runtime (during the overall attack process) of the proposed RL method and baseline methods is needed.
- The distribution learning becomes harder as the dimension becomes larger due to the curse of dimension. Hence, It seems difficult to apply the proposed method to datasets of large dimensions such as ImageNet. If possible, I want to see the experimental results on larger datasets.  If there is not enough time to do this, I would like to see the performance of distribution learning in a large dataset.
- I think a related work section can help readers to understand the difference from the previous works and clarify the contributions of the paper.


POST REBUTTAL COMMENTS:  The authors answer the questions. My concerns have been addressed. They prove that they can apply their method to the larger datasets such as CIFAR-10. Also, they show that their proposed method without online search which has comparable performance in Krum is fast enough. Moreover, the proposed method with online search is fast enough to be executed in a real-world setting. So I decided to adjust my score from 5 to 6.

**Limitations:**

I wish that the authors include ethics or broader-impact statement in the paper or appendix.

**Strengths And Weaknesses:**

# Strengths
- The experimental results on MNIST and EMNIST under Krum and Clipping Median show that the proposed RL-based attack method clearly outperforms the baseline methods.
- The authors theoretically analyze the effect of inaccurate distribution learning and explain how the attack performance loss depends on the ratio of benign workers, the inaccuracy of learned distribution.
- The proposed attack method require less knowledge compared to existing attack methods on FL domain.
- They suggest a state and action space compression technique to make the method scalable.

# Weaknesses
- Lack of some important ablation studies such as actual runtime comparison.
- It seems difficult to apply this method to datasets of large dimension such as ImageNet. Experiments were also conducted on small datasets such as MNIST and EMNIST. In other words, applications of the method seem limited.
- There is no related work section.

# Other Comments
- typo) line 128 A^{\tau(r)} )

---

> ### Author Response · Authors · 2022-08-02
> **Response to Reviewer u8VB**
>
> We appreciate your constructive comments and feedback. Now we will explain your concerns and questions point by point in the following.
>
> Q1: Actual runtime comparison.
>
> A1: The actual runtime varies across the FL environment, the training method used, and most importantly, the amount of computational resource available. The tables below report the numbers from our current experiment settings (see Appendix D.1) and the way the simulator is implemented (clients are simulated sequentially in each FL epoch).
>
> For MNIST and Fashion-MNIST, distribution learning takes around 100 seconds to reconstruct a batch of 32 images and we construct 50 batches within 2 hours. Note that multiple batches can be generated from a single gradient. We start policy training from the beginning of FL training, and we set 8 hours limit for policy training. It takes around 0.05 seconds to simulate a single FL epoch with 10 sampled clients without parallelization. Total training steps vary across defense policies as stated in the supplementary materials D.1.
>
> With the above numbers, if we assume that each FL epoch takes 72 seconds to finish and there are in total of 1,000 FL epochs during FL training, then distribution learning will end before the 100th FL epoch and policy training ends by the 400th FL epochs, and the total FL training time is 20 hours. Once equipped with more powerful devices, the training time can be significantly reduced by parallelly simulating multiple clients using multiprocessing and multiple episodes using vectorized environments, which will make it possible to simulate large FL systems.
>
> In terms of executing time, for MNIST with clipping median defense, IPM takes around 0.25 seconds to execute an attack in each FL epoch, LMP takes around 7.7 seconds, EB takes around 0.5 seconds. The execution time of our RL method varies over the action space used and it takes around 5.8 seconds with the current action space. Given that each FL epoch typically lasts a minute or longer (72 seconds in our experiment), a few seconds of search time is completely acceptable. We observe that for defenses such as Krum, it suffices to use the gradients of the last two layers of model parameters as the action. This approach does not require any online searching and decreases the attack execution time to 0.5s.
>
>                            FL Epochs   Real Time
>      Distribution Learning   100      <= 2 hours
>      Policy Learning         400      <= 8 hours
>      Total FL Training       1000      20 hours
>
>
>                                       Real Executing Time
>       IPM                                   0.25s
>       LMP                                   7.7s
>       EB                                    0.5s
>       RL (with online search)               5.8s
>       RL (without online search)            0.5s
>
>
>
> Q2: CIFAR-10 results
>
> A2: We agree that it is important to understand how our approach works for colored images. From our limited exploration, we found that it is possible to recover a batch of 4 images from CIFAR-10 using the method of Inverting Gradients, which is not very effective for the purpose of distribution learning. However, using gradient leakage to recover training images is a growing area and there are more works trying to recover a large batch of images. For example, GradInversion [2] can recover data from a larger batch (8-48 images) of ImageNet data for ResNets. We will experiment with their approach once the code is available. On the other hand, since we consider an insider attack in this work, the attackers’ local data can be used to build the simulator even without distribution learning. The table below shows that when the attackers use 500 real images from CIFAR-10 (<1% of total data) owned by themselves to train a policy, our RL based method still outperforms other baseline attacks.
>
> CIFAR10 Clipping Median
>
>
>                             200 FL Epochs         600 FL Epochs         1000 FL Epochs
>             No Attack          35.38%                45.38%                 53.7%
>             IPM                28.83%                36.85%                 42.98%
>             EB                 31.96%                43.45%                 10%
>             LMP                14.53%                31.73%                 10%
>             Our Method         10.05%                10.28%                 9.52%
>
>
>
> Q3: Broader impact discussion
>
> A3:  We discussed the broader impact of our work in Appendix A.1. We will provide a detailed discussion of related work in the revised paper.
>
> [1] Huang, Yangsibo, et al. "Evaluating gradient inversion attacks and defenses in federated learning." NeurIPS 2021.
>
> [2] Yin, Hongxu et al. “See through Gradients: Image Batch Recovery via GradInversion.” CVPR 2021

---

> > ### Author Response · Authors · 2022-08-08
> > **Response to Reviewer u8VB**
> >
> > We are glad that we were able to address your concerns. We would like to clarify one comment further regarding the runtime comparison.
> >
> > For Q1: Actual runtime comparison.
> > Since the three stages of our attack overlap with each other, we only need to compare the attack execution time with other baselines (see the second table in Q1). Given that each FL epoch typically lasts a few minutes or longer in practice (due to the necessity of waiting for responses from multiple clients with varying speeds), a few seconds of search time is completely acceptable.

---

> > > ### Author Response · Authors · 2022-08-09
> > > **Thank you for updating your review**
> > >
> > > We are pleased that we have addressed your concerns. We appreciate the updating score and the thorough review.

---

### Official Review · Reviewer_jqbg · 2022-07-22

**Rating:** 6
**Confidence:** 3
**Soundness:** 2 fair
**Presentation:** 1 poor
**Contribution:** 3 good

**Summary:**

The paper studies the vulnerability of Federated Learning (FL) Algorithm for untargeted poisoning attacks. In this direction, it formulates the problem of poisoning FL systems as a model-based Reinforcement Learning.

The RL based attack consists of three key steps: learning the distribution of the data, learning the optimal attack policy by building a simulator from the learned distribution, and finally executing the attack policy. The work presents an upper bound on the attack performance as a function of fraction of malicious nodes in the FL setting and learning rate of the FL algorithm.


**Questions:**

Please see the weaknesses associated with the comparison with the literature, and address them.

**Limitations:**

Yes

**Strengths And Weaknesses:**

The paper provides a three-phased framework for poisoning attacks, however, the novelty in each component is limited. The methods used are previously known and studied in the literature.

Since the problem is modeled as RL, the paper does not compare the poisoning attack proposed in the paper with the poisoning attack proposed in the online RL setting. A similar comparison with the corresponding attack cost in terms of theoretical guarantees is also missing.

This paper is also closely related to the misspecification of distribution of a prior in a Bayesian setting. This line of work is independently studied. Authors should consider a comparison and a literature survey in that direction, which is currently missing.

Overall, the writing of the paper is unclear, and the contributions of the paper are limited.

---

> ### Author Response · Authors · 2022-08-02
> **Response to Reviewer jqbg**
>
> Thank you for your thoughtful and valuable comments on our paper and we will explain your concerns point by point in the following.
>
> Q1: The novelty of the proposed method is limited.
>
> A1: We proposed a novel framework for online attacks against federated learning. Our work shows that it is feasible and beneficial to first learn a world model of the environment and then derive strong attacks from it. Although we focus on model poisoning attacks in this work, our approach makes it easier to incorporate a variety of attack objectives and constraints.
>
> Our distribution learning component borrows concepts from gradient leakage-based inference attacks, but it is the first one (to the best of our knowledge) that learns a data distribution instead of exact data samples. The difference is that some inaccurate data (e.g., wrong labels, noise pixels) is allowed in the learned dataset, as long as they do not dramatically harm the accuracy of the learned distribution, in terms of its distance from the true data distribution (Theorem 1). This explains why we can learn a relatively large batch of data at each epoch and denoise them without reconstructing the exact same images the clients used.
>
> Our model-based approach is different from standard model-based RL methods that explicitly model the transition and the reward functions. Instead, we simulate the FL environment using the learned data distribution. Our three-phased framework provides a practical (where attackers only know their own data), efficient (three phases overlapping with each other), and effective (outperforming state-of-art methods) attack against FL. The framework can be extended to multi-agent cooperative and non-cooperative settings (e.g., for defense purposes), and can be used in other attack settings, such as backdoor attacks (targeted attacks).
>
> Q2: Comparison between online reinforcement learning poisoning attack and our methods
>
> A2: We consider an RL-based attack against federated learning (FL) in this work, which is very different from the problem setting considered in online RL poisoning attacks. The latter line of work studies how to poison an RL agent by perturbing its state or reward signals, but the attack method used does not have to be RL-based. In the FL setting, it is nearly impossible for the attackers to collect enough samples to sufficiently train a complex attack policy (e.g., using a high dimensional neural network), which is typically needed to break a strong defense. Previous online-RL works either ignore sample efficiency or focus on attacks in the testing stage (our attack execution stage), where they assume that the attack policy is already sufficiently trained. In contrast, we solve this problem by simulating environments using the learned distribution, where we can generate sufficient samples, and parallelly run multiple environments (no communication overhead in simulation) when the FL is ongoing. Further, existing RL poisoning attacks have mainly focused on attacking a single RL agent by an external agent rather than an insider attack in a distributed learning environment as we consider. We have briefly mentioned these differences in the introduction section and will expand the discussion in the revised paper.
> Given that the two lines of research consider very different settings, it is difficult to compare them quantitatively.
>
> Q3: Comparison between Bayesian method and our methods
>
> A3: Thanks for the great suggestion. We didn’t apply a Bayesian method to distribution learning in this work because of its complexity when applied to high dimensional data. Instead, we pool together the attackers’ local data and the data generated from distribution learning and use that to generate the attacker’s MDP. We agree that a Bayesian method can potentially boost the accuracy of the learned distribution, although this will also incur additional overhead in the distribution learning stage. Further, it can be integrated with Bayesian RL or distributionally robust RL to improve attack performance. We will provide a detailed survey in this direction as suggested by the reviewer.
>
> [1] Zhang, Xuezhou et al. "Online Data Poisoning Attacks." Proceedings of the 2nd Conference on Learning for Dynamics and Control, 2020.
>
> [2] Zhang, Xuezhou et al. "Adaptive Reward-Poisoning Attacks against Reinforcement Learning." ICML 2020.

---

> > ### Author Response · Authors · 2022-08-09
> > **Response to Reviewer jqbg**
> >
> > We hope that we have addressed your major concerns. If there are still concerns or questions, we would be happy to hear and discuss them. We would like to highlight our contributions to federated learning security by developing novel poisoning attacks in challenging settings. Our RL-based attack not only achieves the SOTA attack performance but also surpasses other baselines by a large gap, especially under strong defenses (e.g., Clipping-Median, Geometric Median [4], and FLTrust [3]). Note that no such attack was known before our work. In particular, previous works have either considered Median [1] or the norm clipping [2] defense but not the combination of the two, and no effective attack against FLTrust [3] was known.
> >
> > For Geometric Median results, please see our response to Reviewer Gb3o Q3.
> >
> > [1] Yin, Dong, et al. "Byzantine-robust distributed learning: Towards optimal statistical rates." International Conference on Machine Learning. PMLR, 2018.
> >
> > [2] Sun, Ziteng, et al. "Can you really backdoor federated learning?" arXiv preprint arXiv:1911.07963 (2019).
> >
> > [3] Cao, Xiaoyu, et al. "Fltrust: Byzantine-robust federated learning via trust bootstrapping." arXiv preprint arXiv:2012.13995 (2020).
> >
> > [4] Pillutla, Krishna, Sham M. Kakade, and Zaid Harchaoui. "Robust aggregation for federated learning." arXiv preprint arXiv:1912.13445 (2019).

---

### Meta-Review · Area_Chair_Ni9X · 2022-08-23

**Recommendation:** Accept
**Confidence:** Certain

**Metareview:**

The recommendation is based on the reviewers' comments, the area chair's personal evaluation, and the post-rebuttal discussion.

This paper proposed a model-based reinforcement learning framework for data poisoning attacks on federated learning. All reviewers find the results convincing and valuable. The authors' rebuttal has successfully addressed the reviewers' concerns. Given the unilateral agreement, I am recommending acceptance

**Award:**

No

---

### Decision · Program_Chairs · 2022-09-14

Accept